# Topographic VAEs learn Equivariant Capsules

**T. Anderson Keller**
UvA-Bosch Delta Lab
University of Amsterdam
`t.anderson.keller@gmail.com`

**Max Welling**
UvA-Bosch Delta Lab
University of Amsterdam
`m.welling@uva.nl`

## Abstract

In this work we seek to bridge the concepts of topographic organization and equivariance in neural networks. To accomplish this, we introduce the Topographic VAE: a novel method for efficiently training deep generative models with topographically organized latent variables. We show that such a model indeed learns to organize its activations according to salient characteristics such as digit class, width, and style on MNIST. Furthermore, through topographic organization over time (i.e. temporal coherence), we demonstrate how predefined latent space transformation operators can be encouraged for observed transformed input sequences – a primitive form of unsupervised learned equivariance. We demonstrate that this model successfully learns sets of approximately equivariant features (i.e. "capsules") directly from sequences and achieves higher likelihood on correspondingly transforming test sequences. Equivariance is verified quantitatively by measuring the approximate commutativity of the inference network and the sequence transformations. Finally, we demonstrate approximate equivariance to complex transformations, expanding upon the capabilities of existing group equivariant neural networks.

## 1 Introduction

Many parts of the brain are organized topographically. Famous examples are the ocular dominance maps and the orientation maps in V1. What is the advantage of such organization and what can we learn from it to develop better inductive biases for deep neural network architectures?

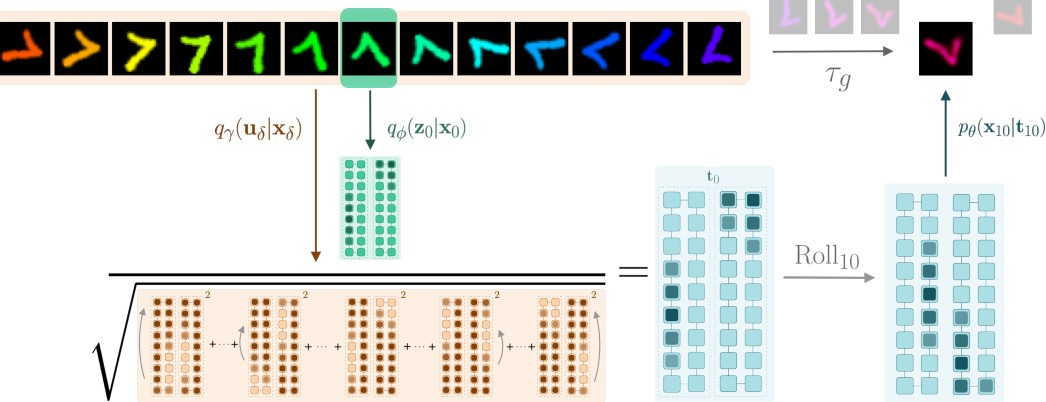

Figure 1: Overview of the Topographic VAE with shifting temporal coherence. The combined color/rotation transformation in input space $\tau_g$ becomes encoded as a Roll within the capsule dimension. The model is thus able decode unseen sequence elements by encoding a partial sequence and Rolling activations within the capsules. We see this resembles a commutative diagram.

35th Conference on Neural Information Processing Systems (NeurIPS 2021).

One potential explanation for the emergence of topographic organization is provided by the principle of redundancy reduction [1]. In the language of Information Theory, redundancy wastes channel capacity, and thus to represent information as efficiently as possible, the brain may strive to transform the input to a neural code where the activations are statistically maximally independent. In the machine learning literature, this idea resulted in Independent Component Analysis (ICA) which linearly transforms the input to a new basis where the activities are independent and sparse [2, 13, 29, 44]. It was soon realized that there are remaining higher order dependencies (such as correlation between absolute values) that can not be transformed away by a linear transformation. For example, along edges of an image, linear-ICA components (e.g. gabor filters) still activate in clusters even though the sign of their activity is unpredictable [48, 56]. This led to new algorithms that explicitly model these remaining dependencies through a topographic organization of feature activations [27, 45, 46, 59]. Such topographic features were reminiscent of pinwheel structures observed in V1, encouraging multiple comparisons with topographic organization in the biological visual system [28, 30, 42].

A second, almost independent body of literature developed the idea of "equivariance" of neural network feature maps under symmetry transformations. The idea of equivariance is that symmetry transformations define equivalence classes as the orbits of their transformations, and we wish to maintain this structure in the deeper layers of a neural network. For instance, for images, asserting a rotated image contains the same object for all rotations, the transformation of rotation then defines an orbit where the elements of that orbit can be interpreted as pose or angular orientation. When an image is processed by a neural network, we want features at different orientations to be able to be combined to form new features, but we want to ensure the relative pose information between the features is preserved for all orientations. This has the advantage that the equivalence class of rotations for the complex composite features is guaranteed to be maintained, allowing for the extraction of invariant features, a unified pose, and increased data efficiency. Such ideas are reminiscent of the capsule networks of Hinton et al. [21, 22, 51], and indeed formal connections to equivariance have been made [39]. Interestingly, by explicitly building neural networks to be equivariant, we additionally see geometric organization of activations into these equivalence classes, and further, the elements within an equivalence class are seen to exhibit higher-order non-Gaussian dependencies [40, 41, 56, 57]. The insight of this connection between topographic organization and equivariance hints at a possibility to encourage approximate equivariance from an induced topology in feature space.

To build a model, we need to ask what mechanisms could induce topographic organization of *observed transformations* specifically? We have argued that removing dependencies between latent variables is a possible mechanism; however, to obtain the more structured organisation of equivariant capsule representations, the usual approach is to hard-code this structure into the network, or to encourage it through regularization terms [4, 15]. To achieve this same structure *unsupervised*, we propose to incorporate another key inductive bias: "temporal coherence" [18, 24, 52, 60]. The principle of temporal coherence, or "slowness", asserts than when processing correlated sequences, we wish for our representations to change smoothly and slowly over space and time. Thinking of time sequences as symmetry transformations on the input, we desire features undergoing such transformations to be grouped into equivariant capsules. We therefore suggest that encouraging slow feature transformations to take place *within a capsule* could induce such grouping from sequences alone.

In the following sections we will explain the details of our Topographic Variational Autoencoder which lies at the intersection of topographic organization, equivariance, and temporal coherence, thereby learning approximately equivariant capsules from sequence data completely unsupervised.

## 2 Related Work

The history of statistical models upon which this work builds is vast, including sparse coding [44], Independant Component Analysis (ICA) [2, 13, 29], Slow Feature Analysis (SFA) [54, 60], and Gaussian scale mixtures [41, 48, 56, 57]. Most related to this work are topographic generative models including Generative Topographic Maps [6], Bubbles [25], Topographic ICA [27], and the Topographic Product of Student's-t [46, 59]. Prior work on learning equivariant and invariant representations is similarly vast and also has a deep relationship with these generative models. Specifically, Independant Subspace Analysis [26, 53], models involving temporal coherence [18, 24, 52, 60], and Adaptive Subspace Self Organizing Maps [35] have all demonstrated the ability to learn invariant feature subspaces and even 'disentangle' space and time [19, 53]. Our work assumes a similar generative model to these works while additionally allowing for efficient estimation of

the model through variational inference [33, 50]. Although our work is not the first to combine Student's-t distributions and variational inference [7], it is the first to provide an efficient method to do so for Topographic Student's-t distributions.

Another line of work has focused on constructing neural networks with equivariant representations separate from the framework of generative modeling. Analytically equivariant networks such as Group Equivariant Neural Networks [11], and other extensions [9, 16, 17, 55, 49, 58, 61, 62] propose to explicitly enforce symmetry to group transformations in neural networks through structured weight sharing. Alternatively, others propose supervised and self-supervised methods for *learning* equivariance or invariance directly from the data itself [4, 14, 15]. One related example in this category uses a group sparsity regularization term to similarly learn topographic features for the purpose of modeling invariance [31]. We believe the Topographic Variational Autoencoder presented in this paper is another promising step in the direction of learning approximate equivariance, and may even hint at how such structure could be learned in biological neural networks.

Furthermore, the idea of disentangled representations [3] has also been been connected to equivariance and representation theory in multiple recent papers [8, 12, 10, 20]. Our work shares a fundamental connection to this distributed operator definition of disentanglement, where the slow roll of capsule activations can be seen as the latent operator. Recently, the authors of [34] demonstrated that incorporating the principle of 'slowness' in a variational autoencoder (VAE) yields the ability to learn disentangled representations from natural sequences. While similar in motivation, the generative model proposed in [34] is unrelated to topographic organization and equivariance, and is more aligned with traditional notions of disentanglement.

Finally, and importantly, in the neuroscience literature, another popular explanation for topographic organization arises as the solution to the 'wiring length' minimization problem [36]. Recently, models which attempt to incorporate wiring length constraints have been shown to yield topographic organization of higher level features, ultimately resembling the 'face patches' found in primates [32, 38]. Interestingly, the model presented in this paper organizes activity based on the same statistical property (local correlation) as the wiring length proxies developed in [38], but from a generative modeling perspective, demonstrating a computationally principled explanation for the same phenomenon.

## 3 Background

The model in this paper is a first attempt at bridging two yet disjoint classes of models: Topographic Generative Models, and Equivariant Neural Networks. In this section, we will provide a brief background on these two frameworks.

### 3.1 Topographic Generative models

Inspired by Topographic ICA, the class of Topographic Generative models can be understood as generative models where the joint distribution over latent variables does not factorize into entirely independent factors, as is commonly done in ICA or VAEs, but instead has a more complex 'local' correlation structure. The locality is defined by arranging the latent variables into an n-dimensional lattice or grid, and organizing variables such that those which are closer together on this grid have greater correlation of activities than those which are further apart. In the related literature, activations which are nearby in this grid are defined to have higher-order correlation, e.g. correlations of squared activations (aka 'energy'), asserting that all first order correlations are removed by the initial ICA de-mixing matrix.

Such generative models can be seen as hierarchical generative models where there exist higher level independent 'variance generating' variables $\mathbf{V}$ which are combined locally to generate the variances $\boldsymbol{\sigma} = \phi(\mathbf{WV})$ of the lower level topographic variables $\mathbf{T} \sim \mathcal{N}(\mathbf{0}, \boldsymbol{\sigma}^2\mathbf{I})$, for an appropriate non-linearity $\phi$. The variables $\mathbf{T}$ are thus independent conditioned on $\boldsymbol{\sigma}$. Other related models which can be described under this umbrella include *Independent Subspace Analysis* (ISA) [26] where all variables within a predefined subspace (or 'capsule') share a common variance, and '*temporally coherent*' models [24] where the energy of a given variable between time steps is correlated by extending the topographic neighborhoods over the time dimension [25]. The topographic latent variable $\mathbf{T}$ can additionally be described as an instance of a Gaussian scale mixture (GSM). GSMs have previously been used to model the observed non-Gaussian dependencies between coefficients of steerable wavelet pyramids (interestingly also equivariant to translation & rotation) [48, 56, 57].

## 3.2 Group Equivariant Neural Networks

Equivariance is the mathematical notion of symmetry for functions. A function is said to be an equivariant map if the the result of transforming the input and then computing the function is the same as first computing the function and then transforming the output. In other words, the function and the transformation commute. Formally, $f(\tau_\rho[\mathbf{x}]) = \Gamma_\rho[f(\mathbf{x})]$, where $\tau$ and $\Gamma$ denote the (potentially different) operators on the domain and co-domain respectively, but are indexed by the same element $\rho$.

It is well known that convolutional maps in neural networks are translation equivariant, i.e., given a translation $\Gamma_\rho$ (applied to each feature map separately) and a convolutional map $f(\cdot)$, we have $f(\Gamma_\rho[\mathbf{x}]) = \Gamma_\rho[f(\mathbf{x})]$. This can be extended to other transformations (e.g. rotation or mirroring) using Group convolutions ($G$-convolutions) [11]. As a result of the design of $G$-convolutions, feature maps that are related to each other by a rotation of the filter/input are grouped together. Moreover, a rotation of the input results in a transformation (i.e. a permutation and rotation) on the activations of each of these groups in the output. Hence, we can think of these equivalence class groups as capsules where transformations of the input only cause structured transformations *within* a capsule. As we will demonstrate later, this is indeed analogous to the structure of the representation learned by the Topographic VAE with temporal coherence – a transformation of the input yields a cyclic permutation of activations *within* each capsule. However, due to the approximate *learned* nature of the equivariant representation, the Topographic VAE does not require the transformations $\tau_\rho$ to constitute a group.

# 4 The Generative Model

The generative model proposed in this paper is based on the Topographic Product of Student's-t (TPoT) model as developed in [46, 59]. In the following, we will show how a TPoT random variable can be constructed from a set of independent univariate standard normal random variables, enabling efficient training through variational inference. Subsequently, we will construct a new model where topographic neighborhoods are extended over time, introducing temporal coherence and encouraging the unsupervised learning of approximately equivariant subspaces we call 'capsules'.

## 4.1 The Product of Student's-t Model

We assume that that our observed data is generated by a latent variable model where the joint distribution over observed and latent variables $\mathbf{x}$ and $\mathbf{t}$ factorizes into the product of the conditional and the prior. The prior distribution $p_{\mathbf{T}}(\mathbf{t})$ is assumed to be a Topographic Product of Student's-t (TPoT) distribution, and we parameterize the conditional distribution with a flexible function approximator:

$$p_{\mathbf{X},\mathbf{T}}(\mathbf{x},\mathbf{t}) = p_{\mathbf{X}|\mathbf{T}}(\mathbf{x}|\mathbf{t})p_{\mathbf{T}}(\mathbf{t}) \qquad p_{\mathbf{X}|\mathbf{T}}(\mathbf{x}|\mathbf{t}) = p_\theta(\mathbf{x}|g_\theta(\mathbf{t})) \qquad p_{\mathbf{T}}(\mathbf{t}) = \text{TPoT}(\mathbf{t};\nu) \qquad (1)$$

The goal of training is thus to learn the parameters $\theta$ such that the marginal distribution of the model $p_\theta(\mathbf{x})$ matches that of the observed data. Unfortunately, the marginal likelihood is generally intractable except for all but the simplest choices of $g_\theta$ and $p_{\mathbf{T}}$ [45]. Prior work has therefore resorted to techniques such as contrastive divergence with Gibbs sampling [59] to train TPoT models as energy based models. In the following section, we instead demonstrate how TPoT variables can be constructed as a deterministic function of Gaussian random variables, enabling the use of variational inference and efficient maximization of the likelihood through the evidence lower bound (ELBO).

## 4.2 Constructing the Product of Student's-t Distribution

First, note a univariate Student's-t random variable $T$ with $\nu$ degrees of freedom can be defined as:

$$T = \frac{Z}{\sqrt{\frac{1}{\nu}\sum_i^\nu U_i^2}} \quad \text{with} \quad Z, U_i \sim \mathcal{N}(0,1) \ \forall i \qquad (2)$$

Where $Z$ and $\{U_i\}_{i=1}^\nu$ are independent standard normal random variables. If $\mathbf{T}$ is a multidimensional Student's-t random variable, composed of independent $Z_i$ and $U_i$, then $\mathbf{T} \sim \text{PoT}(\nu)$, i.e.:

$$\mathbf{T} = \left[ \frac{Z_1}{\sqrt{\frac{1}{\nu}\sum_{i=1}^\nu U_i^2}}, \ \frac{Z_2}{\sqrt{\frac{1}{\nu}\sum_{i=\nu+1}^{2\cdot\nu} U_i^2}}, \ \cdots \ \frac{Z_n}{\sqrt{\frac{1}{\nu}\sum_{i=(n-1)\cdot\nu+1}^{n\cdot\nu} U_i^2}} \right] \sim \text{PoT}(\nu) \qquad (3)$$

Note that the Student's-t variable $T$ is large when most of the $\{U_i\}_i$ in its set are small. We can therefore think of the $\{U_i\}_i$ as constraint violations rather then pattern matches: if the input matches all constraints $U_i \approx 0$, the corresponding $T$ variables will activate (see [23] for further discussion).

### 4.3 Introducing Topography

To make the PoT distribution topographic, we strive to correlate the scales of $T_j$ which are 'nearby' in our topographic layout. One way to accomplish this is by *sharing* some $U_i$-variables between neighboring $T_j$'s. Formally, we define overlapping neighborhoods $\mathsf{N}(j)$ for each variable $T_j$ and write:

$$\mathbf{T} = \left[ \frac{Z_1}{\sqrt{\frac{1}{\nu}\sum_{i\in\mathsf{N}(1)} U_i^2}}, \ \frac{Z_2}{\sqrt{\frac{1}{\nu}\sum_{i\in\mathsf{N}(2)} U_i^2}}, \ \cdots \ \frac{Z_n}{\sqrt{\frac{1}{\nu}\sum_{i\in\mathsf{N}(n)} U_i^2}} \right] \sim \mathrm{TPoT}(\nu) \qquad (4)$$

With some abuse of notation, if we define $\mathbf{W}$ to be the adjacency matrix which defines our neighborhood structure, $\mathbf{U}$ and $\mathbf{Z}$ to be the vectors of random variables $U_i$ and $Z_j$, we can write the above succinctly as:

$$\mathbf{T} = \left[ \frac{Z_1}{\sqrt{\frac{1}{\nu}W_1\mathbf{U}^2}}, \ \frac{Z_2}{\sqrt{\frac{1}{\nu}W_2\mathbf{U}^2}}, \ \cdots \ \frac{Z_n}{\sqrt{\frac{1}{\nu}W_n\mathbf{U}^2}} \right] = \frac{\mathbf{Z}}{\sqrt{\frac{1}{\nu}\mathbf{W}\mathbf{U}^2}} \sim \mathrm{TPoT}(\nu) \qquad (5)$$

Due to non-linearities such as ReLUs which may alter input distributions, it is beneficial to allow the $Z$ variables to model the mean and scale. We found this can be achieved with the following parameterization: $\mathbf{T} = \frac{\mathbf{Z}-\mu}{\sigma\sqrt{1/\nu\mathbf{W}\mathbf{U}^2}}$. In practice, we found that $\sigma = \sqrt{\nu}$ often works well, finally yielding:

$$\mathbf{T} = \frac{\mathbf{Z}-\mu}{\sqrt{\mathbf{W}\mathbf{U}^2}} \qquad (6)$$

Given this construction, we observe that the TPoT generative model can instead be viewed as a latent variable model where all random variables are Gaussian and the construction of $\mathbf{T}$ in Equation 6 is the first layer of the generative 'decoder': $g_\theta(\mathbf{t}) = g_\theta(\mathbf{u}, \mathbf{z})$. In Section 5 we then leverage this interpretation to show how an approximate posterior for the latent variables $\mathbf{Z}$ and $\mathbf{U}$ can be trained through variational inference.

### 4.4 Capsules as Disjoint Topologies

One setting of neighborhood structure $\mathbf{W}$ which is of particular interest is when there exist multiple sets of disjoint neighborhoods. Statistically, the variables of two disjoint topologies are completely independent. An example of a capsule neighborhood structure is shown in Figure 2. The idea of independant subspaces has previously been shown to learn invariant feature subspaces in the linear setting and is present in early work on Independent Subspace Analysis [26] and Adaptive Subspace Self Organizing Maps (AS-SOM) [35]. It is also very reminiscent of the transformed sets of features present in a group equivariant convolutional neural network. In the next section, we will show how temporal coherence can be leveraged to induce the encoding of observed transformations into the internal dimensions of such capsules thereby yielding unsupervised approximately equivariant capsules.

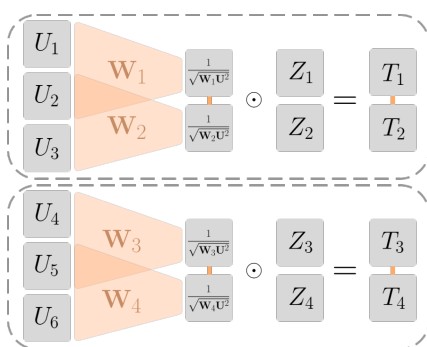

Figure 2: An example of a neighborhood structure which induces disjoint topologies (aka capsules). Lines between variables $T_i$ indicate that sharing of $U_i$, and thus correlation.

### 4.5 Temporal Coherence and Learned Equivariance

We now describe how the induced topographic organization can be leveraged to learn a basis of approximately equivariant capsules for observed transformation sequences. The resulting representation is composed of a large set of 'capsules' where the dimensions inside the capsule are topographically structured, but between the capsules there is independence. To benefit from sequences of input, we encourage topographic structure over time between sequentially permuted activations within a capsule, a property we refer to as *shifting temporal coherence*.

#### 4.5.1 Temporal Coherence

Temporal Coherence can be measured as the correlation of squared activation between time steps. One way we can achieve this in our model is by having $T_j$ share $U_i$ between time steps. Formally, the generative model is identical to Equation 1, factorizing over timesteps denoted by subscript $l$, i.e. $p_{\mathbf{X}_l, \mathbf{T}_l}(\mathbf{x}_l, \mathbf{t}_l) = p_{\mathbf{X}_l|\mathbf{T}_l}(\mathbf{x}_l|\mathbf{t}_l)p_{\mathbf{T}_l}(\mathbf{t}_l)$. However, $\mathbf{T}_l$ is now a function of a sequence $\{\mathbf{U}_{l+\delta}\}_{\delta=-L}^{L}$:

$$\mathbf{T}_l = \frac{\mathbf{Z}_l - \mu}{\sqrt{\mathbf{W}\left[\mathbf{U}_{l+L}^2; \cdots; \mathbf{U}_{l-L}^2\right]}} \tag{7}$$

Where $\left[\mathbf{U}_{l+L}^2; \cdots; \mathbf{U}_{l-L}^2\right]$ denotes vertical concatenation of the column vectors $\mathbf{U}_l$, and $2L$ can be seen as the window size. We see that the choice of $\mathbf{W}$ now defines correlation structure over time. In prior work on temporal coherence (denoted 'Bubbles' [25]), the grouping over time is such that a given variable $T_{l,i}$ has correlated energy with *the same spatial location* $(i)$ at a previous time step $(l-1)$ $\left(\text{i.e. } \text{cov}(T_{l,i}^2, T_{l-1,i}^2) > 0\right)$. This can be implemented as:

$$\mathbf{W}\left[\mathbf{U}_{l+L}^2; \cdots; \mathbf{U}_{l-L}^2\right] = \sum_{\delta=-L}^{L} \mathbf{W}_\delta \mathbf{U}_{l+\delta}^2 \tag{8}$$

Where $\mathbf{W}_\delta$ defines the topography for a single timestep, and is typically the same for all timesteps.

#### 4.5.2 Learned Equivariance with Shifting Temporal Coherence

In our model, instead of requiring a single location to have correlated energies over a sequence, we would like variables at sequentially permuted locations *within a capsule* to have correlated energy between timesteps $\left(\text{cov}(T_{l,i}^2, T_{l-1,i-1}^2) > 0\right)$. Similarly, this can be implemented as:

$$\mathbf{W}\left[\mathbf{U}_{l+L}^2; \cdots; \mathbf{U}_{l-L}^2\right] = \sum_{\delta=-L}^{L} \mathbf{W}_\delta \text{Roll}_\delta(\mathbf{U}_{l+\delta}^2) \tag{9}$$

Where $\text{Roll}_\delta(\mathbf{U}_{l+\delta}^2)$ denotes a cyclic permutation of $\delta$ steps along the capsule dimension. The exact implementation of Roll can be found in Section A.11. As we will show in Section 6.3, TVAE models with such a topographic structure learn to encode observed sequence transformations as Rolls within the capsule dimension, analogous to a group equivariant neural network where $\tau_\rho$ and $\text{Roll}_1$ can be seen as the action of the transformation $\rho$ on the input and output spaces respectively.

## 5 Topographic VAE

To train the parameters of the generative model $\theta$, we use the above formulation to parameterize an approximate posterior for $\mathbf{t}$ in terms of a deterministic transformation of approximate posteriors over simpler Gaussian latent variables $\mathbf{u}$ and $\mathbf{z}$. Explicitly:

$$q_\phi(\mathbf{z}_l|\mathbf{x}_l) = \mathcal{N}\big(\mathbf{z}_l; \mu_\phi(\mathbf{x}_l), \sigma_\phi(\mathbf{x}_l)\mathbf{I}\big) \qquad p_\theta(\mathbf{x}_l|g_\theta(\mathbf{t}_l)) = p_\theta(\mathbf{x}_l|g_\theta(\mathbf{z}_l, \{\mathbf{u}_l\})) \tag{10}$$

$$q_\gamma(\mathbf{u}_l|\mathbf{x}_l) = \mathcal{N}\big(\mathbf{u}_l; \mu_\gamma(\mathbf{x}_l), \sigma_\gamma(\mathbf{x}_l)\mathbf{I}\big) \qquad \mathbf{t}_l = \frac{\mathbf{z}_l - \mu}{\sqrt{\mathbf{W}\left[\mathbf{u}_{l+L}^2; \cdots; \mathbf{u}_{l-L}^2\right]}} \tag{11}$$

We denote this model the Topographic VAE (TVAE) and optimize the parameters $\theta, \phi, \gamma$ (and $\mu$) through the ELBO, summed over the sequence length $S$:

$$\sum_{l=1}^{S} \mathbb{E}_{Q_{\phi,\gamma}(\mathbf{z}_l,\mathbf{u}_l|\{\mathbf{x}_l\})} \big([\log p_\theta(\mathbf{x}_l|g_\theta(\mathbf{t}_l))] - D_{KL}[q_\phi(\mathbf{z}_l|\mathbf{x}_l)||p_\mathbf{Z}(\mathbf{z}_l)] - D_{KL}[q_\gamma(\mathbf{u}_l|\mathbf{x}_l)||p_\mathbf{U}(\mathbf{u}_l)]\big) \tag{12}$$

where $Q_{\phi,\gamma}(\mathbf{z}_l, \mathbf{u}_l|\{\mathbf{x}_l\}) = q_\phi(\mathbf{z}_l|\mathbf{x}_l) \prod_{\delta=-L}^{L} q_\gamma(\mathbf{u}_{l+\delta}|\mathbf{x}_{l+\delta})$, and $\{\cdot\}$ denotes a set over time.

## 6 Experiments

In the following experiments, we demonstrate the viability of the Topographic VAE as a novel method for training deep topographic generative models. Additionally, we quantitatively verify that shifting

temporal coherence yields approximately equivariant capsules by computing an 'equivariance loss' and a correlation metric inspired by the disentanglement literature. We show that equivariant capsule models yield higher likelihood than baselines on test sequences, and qualitatively support these results with visualizations of sequences reconstructed purely from Rolled capsule activations.

## 6.1 Evaluation Methods

As depicted in Figure 1, we make use of *capsule traversals* to qualitatively visualize the transformations learned by our network. Simply, these are constructed by encoding a partial sequence into a $\mathbf{t}_0$ variable, and decoding sequentially Rolled copies of this variable. Explicitly, in the top row we show the data sequence $\{\mathbf{x}_l\}_l$, and in the bottom row we show the decoded sequence: $\{g_\theta(\mathrm{Roll}_l(\mathbf{t_0}))\}_l$.

To measure equivariance quantitatively, we measure an *equivariance error* similar to [15]. The equivariance error can be seen as the difference between traversing the two distinct paths of the commutative diagram, and provides some measure of how precisely the function and the transform commute. Formally, for a sequence of length $S$, and $\hat{\mathbf{t}} = \mathbf{t}/||\mathbf{t}||_2$, the error is defined as:

$$\mathcal{E}_{eq}(\{\mathbf{t}_l\}_{l=1}^S) = \sum_{l=1}^{S-1}\sum_{\delta=1}^{S-l} \left|\left|\mathrm{Roll}_\delta(\hat{\mathbf{t}}_l) - \hat{\mathbf{t}}_{l+\delta}\right|\right|_1 \tag{13}$$

Additionally, inspired by existing disentanglement metrics, we measure the degree to which observed transformations in capsule space are correlated with input transformations by introducing a new metric we call $\mathrm{CapCorr}_y$. Simply, this metric computes the correlation between the amount of observed Roll of a capsule's activation at two timesteps $l$ and $l + \delta$, and the shift of the ground truth generative factors $y_l$ in that same time. Formally, for a correlation coefficient $\mathrm{Corr}$:

$$\mathrm{CapCorr}(\mathbf{t}_l, \mathbf{t}_{l+\delta}, y_l, y_{l+\delta}) = \mathrm{Corr}\left(\mathrm{argmax}\left[\mathbf{t}_l \star \mathbf{t}_{l+\delta}\right], |y_l - y_{l+\delta}|\right) \tag{14}$$

Where $\star$ is discrete periodic cross-correlation across the capsule dimension, and the correlation coefficient is computed across the entire dataset. We see the $\mathrm{argmax}$ of the cross-correlation is an estimate of the degree to which a capsule activation has shifted from time $l$ to $l + \delta$. To extend this to multiple capsules, we can replace the $\mathrm{argmax}$ function with the mode of the $\mathrm{argmax}$ computed for all capsules. We provide additional details and extensions of this metric in Section A.10. For measuring capsule-metrics on baseline models which do not naturally have capsules, we simply arbitrarily divide the latent space into a fixed set of corresponding capsules and capsule dimensions, and provide such results as equivalent to 'random baselines' for these metrics.

## 6.2 Topographic VAE without Temporal Coherence

To validate the TVAE is capable of learning topographically organized representations with deep neural networks, we first perform experiments on a Topographic VAE without Temporal Coherence. The model is constructed as in Equations 10 and 11 with $L = 0$, and is trained to maximize Equation 12. We fix $\mathbf{W}$ such that globally the latent variables are arranged in a grid on a 2-dimensional torus (a single capsule), and locally $\mathbf{W}$ sums over 5x5 2D groups of variables. In this setting, $\mathbf{W}$ can be easily implemented as 2D convolution with a 5x5 kernel of 1's, stride 1, and cyclic padding. We see that training the model with 3-layer MLP's for the encoders and decoder indeed yields a 2D topographic organization of higher level features. In Figure 3, we show the maximum activating image for each final layer neuron of the capsule, plotted as a flattened torus. We see that the neurons become arranged according to class, orientation, width, and other learned features.

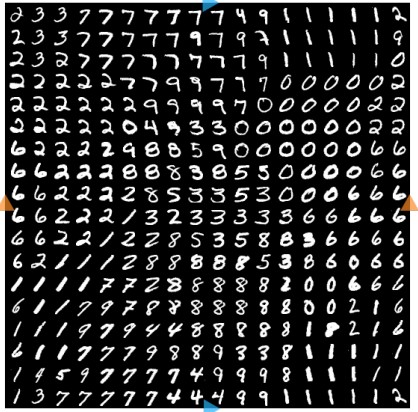

Figure 3: Maximum activating images for a Topographic VAE trained with a 2D torus topography on MNIST.

## 6.3 Learning Equivariant Capsules

In the remaining experiments, we provide evidence that the Topographic VAE can be leveraged to learn equivariant capsules by incorporating shifting temporal coherence into a 1D baseline topographic

model. We compare against two baselines: standard normal VAEs and models that have non-shifting 'stationary' temporal coherence as defined in Equation 8 (denoted 'BubbleVAE' [25]).

In all experiments we use a 3-layer MLP with ReLU activations for both encoders and the decoder. We arrange the latent space into 15 circular capsules each of 15-dimensions for dSprites [43], and 18 circular capsules each of 18-dimensions for MNIST [37]. Example sequences $\{\mathbf{x}_l\}_{l=1}^{S}$ are formed by taking a random initial example, and sequentially transforming it according to one of the available transformations: (X-Pos, Y-Pos, Orientation, Scale) for dSprites, and (Color, Scale, Orientation) for MNIST. All transformation sequences are cyclic such that when the maximum transformation parameter is reached, the subsequent value returns to the minimum. We denote the length of a full transformation sequence by $S$, and the time-extent of the induced temporal coherence (i.e. the length of the input sequence) by $2L$. For simplicity, both datasets are constructed such that the sequence length $S$ equals the capsule dimension (for dSprites this involves taking a subset of the full dataset and looping the scale 3-times for a scale-sequence). Exact details are in Sections A.8 & A.9.

In Figure 4, we show the capsule traversals for TVAE models with $L \approx \frac{1}{3}S$. We see that despite the $\mathbf{t}_0$ variable encoding only $\frac{2}{3}$ of the sequence, the remainder of the transformation sequence can be decoded nearly perfectly by permuting the activation through the full capsule – implying the model has learned to be approximately equivariant to full sequences while only observing partial sequences per training point. Furthermore, we see that the model is able to successfully learn all transformations simultaneously for the respective datasets.

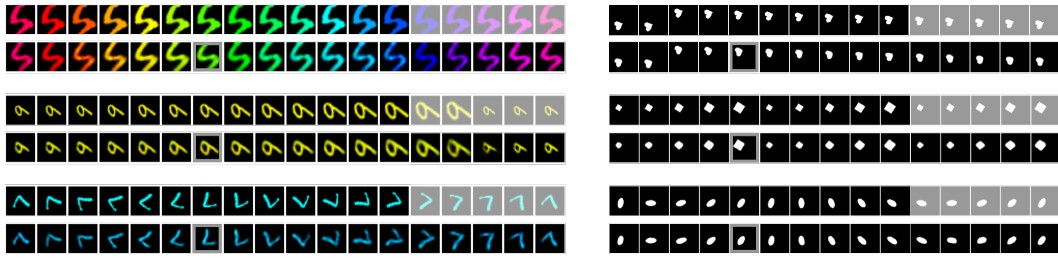

Figure 4: Capsule Traversals for TVAE models on dSprites and MNIST. The top rows show the encoded sequences (with greyed-out images held-out), and the bottom rows show the images generated by decoding sequentially $\mathrm{Rolled}$ copies of the initial activation $\mathbf{t}_0$ (indicated by a grey border).

Capsule traversals for the non-equivariant baselines, as well as TVAEs with smaller values of $L$ (which only learn approximate equivariance to partial sequences) are shown in Section D. We note that the capsule traversal plotted in Figure 1 demonstrates a transformation where color and rotation change simultaneously, differing from how the models in this section are trained. However, as we describe in more detail in Section B.4, we observe that TVAEs trained with individual transformations in isolation (as in this section) are able to generalize, generating sequences of combined transformations when presented with such partial input sequences at test time. We believe this generalization capability to be promising for data efficiency, but leave further exploration to future work. Additional capsule traversals with such unseen combined transformations are shown in Section B.4 and further complex learned transformations (such as perspective transforms) are shown at the end of Section D.

For a more quantitative evaluation, in Table 1 we measure the equivariance error and log-likelihood (reported in nats) of the test data under our trained MNIST models as estimated by importance sampling with 10 samples. We observe that models which incorporate temporal coherence (Bubble-VAE and TVAE with $L > 0$) achieve low equivariance error, while the TVAE models with shifting temporal coherence achieve the highest likelihood and the lowest equivariance error simultaneously.

Table 1: Log Likelihood and Equivariance Error on MNIST for different settings of temporal coherence length $L$ relative to sequence length $S$. Mean $\pm$ std. over 3 random initalizations.

| Model | TVAE | TVAE | TVAE | BubbleVAE | VAE |
|---|---|---|---|---|---|
| $L$ | $L = \frac{1}{2}S$ | $L = \frac{5}{36}S$ | $L = 0$ | $L = \frac{5}{36}S$ | $L = 0$ |
| $\log p(\mathbf{x}) \uparrow$ | $-\mathbf{186.8} \pm 0.1$ | $-\mathbf{186.0} \pm 0.7$ | $-218.5 \pm 0.9$ | $-191.4 \pm 0.5$ | $-189.0 \pm 0.8$ |
| $\mathcal{E}_{eq} \downarrow$ | $\mathbf{574} \pm 2$ | $3247 \pm 3$ | $3217 \pm 105$ | $3370 \pm 12$ | $13274 \pm 1$ |

Table 2: Equivariance error ($\mathcal{E}_{eq} \downarrow$) and correlation of observed capsule roll with ground truth factor shift (CapCorr $\uparrow$) for the dSprites dataset. Mean $\pm$ standard deviation over 3 random initalizations.

| Model
$L$ | TVAE
$L = \frac{1}{2}S$ | TVAE
$L = \frac{1}{3}S$ | TVAE
$L = \frac{1}{6}S$ | TVAE
$L = 0$ | BubbleVAE
$L = \frac{1}{3}S$ | VAE
$L = 0$ |
|---|---|---|---|---|---|---|
| CapCorr$_X \uparrow$ | $\mathbf{1.0 \pm 0}$ | $\mathbf{1.0 \pm 0}$ | $0.67 \pm 0.02$ | $0.17 \pm 0.03$ | $0.13 \pm 0.01$ | $0.18 \pm 0.01$ |
| CapCorr$_Y \uparrow$ | $\mathbf{1.0 \pm 0}$ | $\mathbf{1.0 \pm 0}$ | $0.66 \pm 0.02$ | $0.21 \pm 0.02$ | $0.12 \pm 0.01$ | $0.16 \pm 0.01$ |
| CapCorr$_O \uparrow$ | $\mathbf{1.0 \pm 0}$ | $\mathbf{1.0 \pm 0}$ | $0.52 \pm 0.01$ | $0.09 \pm 0.01$ | $0.10 \pm 0.01$ | $0.11 \pm 0.00$ |
| CapCorr$_S \uparrow$ | $\mathbf{1.0 \pm 0}$ | $\mathbf{1.0 \pm 0}$ | $0.42 \pm 0.01$ | $0.51 \pm 0.01$ | $0.50 \pm 0.00$ | $0.52 \pm 0.00$ |
| $\mathcal{E}_{eq} \downarrow$ | $\mathbf{344 \pm 5}$ | $1034 \pm 6$ | $2549 \pm 38$ | $2971 \pm 9$ | $1951 \pm 34$ | $6934 \pm 0$ |

To further understand how capsules transform for observed input transformations, in Table 2 we measure $\mathcal{E}_{eq}$ and the CapCorr metric on the dSprites dataset for the four proposed transformations. We see that the TVAE with $L \geq \frac{1}{3}S$ achieves perfect correlation – implying the learned representation indeed permutes cyclically within capsules for observed transformation sequences. Further, this correlation gradually decreases as $L$ decreases, eventually reaching the same level as the baselines. We also see that, on both datasets, the equivariance losses for the TVAE with $L = 0$ and the BubbleVAE are significantly lower than the baseline VAE, while conversely, the CapCorr metric is not significantly better. We believe this to be due to the fundamental difference between the metrics: $\mathcal{E}_{eq}$ measures continuous L1 similarity which is still low when a representation is locally smooth (even if the change of the representation does not follow the observed transformation), whereas CapCorr more strictly measures the correspondence between the transformation of the input and the transformation of the representation. In other words, $\mathcal{E}_{eq}$ may be misleadingly low for invariant capsule representations (as with the BubbleVAE), whereas CapCorr strictly measures equivariance.

# 7 Future Work & Limitations

The model presented in this work has a number of limitations in its existing form which we believe to be interesting directions for future research. Foremost, the model is challenging to compare directly with existing disentanglement and equivariance literature since it requires an input sequence which determines the transformations reachable through the capsule roll. Related to this, we note the temporal coherence proposed in our model is not 'causal' (i.e. $\mathbf{t}_0$ depends on future $\mathbf{x}_l$). We believe these limitations could be at least partially alleviated with minor extensions detailed in Section C.

We additionally note that some model developers may find a priori definition of topographic structure burdensome. While true, we know that the construction of appropriate priors is always a challenging task in latent variable models, and we observe that our proposed TVAE achieves strong performance even with improper specification. Furthermore, in future work, we believe adding learned flexibility to the parameters $\mathbf{W}$ may alleviate some of this burden.

Finally, we note that while this work does demonstrate improved log-likelihood and equivariance error, the study is inherently preliminary and does not examine all important benefits of topographic or approximately equivariant representations. Specifically, further study of the TVAE both with and without temporal coherence in terms of the sample complexity, semi-supervised classification accuracy, and invariance through structured topographic pooling would be enlightening.

# 8 Conclusion

In the above work we introduce the Topographic Variational Autoencoder as a method to train deep topographic generative models, and show how topography can be leveraged to learn approximately equivariant sets of features, a.k.a. capsules, directly from sequences of data with no other supervision. Ultimately, we believe these results may shine some light on how biological systems could hard-wire themselves to more effectively learn representations with equivariant capsule structure. In terms of broader impact, it is foreseeable our model could be used to generate more realistic transformations of 'deepfakes', enhancing disinformation. Given that the model learns *approximate* equivariance, we caution against the over-reliance on equivariant properties as these have no known formal guarantees.

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

# 9   Acknowledgements

We would like to thank Jorn Peters for his invaluable contributions to this work at its earliest stages. We would additionally like to thank Patrick Forré, Emiel Hoogeboom, and Anna Khoreva for their helpful guidance throughout the project. We would like to thank the creators of Weight & Biases [5] and PyTorch [47]. Without these tools our work would not have been possible. Finally, we thank the Bosch Center for Artificial Intelligence for funding, and the reviewers for their helpful comments.

