# A   Experiment Details

The code for reproducing all experiments in this paper can be found in the following GitHub repository: `https://github.com/AKAndyKeller/TopographicVAE`

## A.1   Optimizer Parameters

Given the differences between the training procedures of the model presented in Section 6.2, and those in Section 6.3, the optimizer parameters for the two settings differed slightly. The 2D Topographic VAE without Temporal Coherence presented in Figure 3 was trained with stochastic gradient descent on batches of size 128, using a learning rate of $1 \times 10^{-4}$, and standard momentum of 0.9 for 250 epochs. All models in Section 6.3 were trained with stochastic gradient descent on batches of size 8 (due to each batch-example being a length 15 or 18 sequence), using a learning rate of $1 \times 10^{-4}$, and standard momentum of 0.9 for 100 epochs.

## A.2   Initalization

All weights of the models were initialized with uniformly random samples from $U(-\frac{1}{\sqrt{m}}, \frac{1}{\sqrt{m}})$, where $m$ is the number of input units. For all topographic models including BubbleVAE, $\mu$ was initialized to a large value (30.0) as this was observed to increase the speed of convergence and was sometimes necessary for observed topographic organization in deeper models. For the 2D topographic model in Figure 3, $\mu$ was initialized to 10.

## A.3   Model Architectures

All models presented in this paper make use of the same 3-Layer MLP for parameterizing the encoders and decoders. Specifically, the model is constructed as 3 fully connected layers with ReLU activations in-between the layers. For MNIST, the layers of both the $\mathbf{u}$ and $\mathbf{z}$ encoders have (972, 648, 648) output units each for the first, second, and third layers respectively. The 648 units in the third layer are divided into two sets to compute the mean and log standard deviation of the respective $u$'s and $z$'s, yielding 324 $t$ variables. This is then divided into 18 capsules, each of 18 dimensions. The layers of the decoder have (648, 972, 2352) output units respectively. For dSprites, both encoder layers have output sizes (674, 450, 450), where the resulting 225 $t$ variables are divided into 15 capsules, each of 15 dimensions. The decoder layers then have output sizes (450, 675, 4096). We note the non-topographic VAE baselines make use of only a single encoder for the Gaussian variable $\mathbf{z}$ (as $\mathbf{u}$ is not needed), and do not incorporate a $\mu$ parameter.

## A.4   Choice of W

For all topographic models (TVAE and BubbleVAE) in Section 6.3, the global topographic organization afforded by $\mathbf{W}$ was fixed to a set of 1-D tori ('circular capsules') as depicted in Figure 1. The model presented in Section 6.2 organizes its variables as a single 2-D torus. Practically, multiplication by $\mathbf{W}$ was performed by convolution over the appropriate dimensions (time & capsule dimension) with a kernel of all 1's, taking advantage of circular padding to achieve toroidal structure.

## A.5   Choice of $\mathbf{W}_\delta$

The choice of $\mathbf{W}_\delta$ determines the local topographic structure within a single timestep. For all TVAE models with $L > 0$, we experimented with local neighborhood sizes (denoted $K$) of 3 units (effective kernel size 3 in the capsule dimension), and 1 unit (no neighborhood). For MNIST it was observed that $K = 3$ performed best, while $K = 1$ worked best for dSprites. This is likely due to the slower, smoother, and more overlapping transformations constructed on MNIST, whereas our subset of dSprites contained non-smooth transformations where the overlap between successive images was smaller (e.g. due to sub-setting, see Section A.9), which made larger neighborhood sizes $K > 1$ less fitting. For TVAE models with $L = 0$, $\mathbf{W}_\delta = \mathbf{W}$ was fixed to sum over neighborhoods of size $K = 9$ for MNIST and $K = 3$ for dSprites. These values were chosen to be sufficiently large to achieve notably lower equivariance error than the VAE baseline, and thus demonstrate the impact of topographic organization without temporal coherence. For BubbleVAE models, the extent of topographic organization in the capsule dimension was set to $K = 3$ on MNIST to match the TVAE,

and was set to be equal to the organization in time dimension $K = 2L$ for dSprites. A further quantitative comparison on the impact of the choice of the $K$ parameter can be found in Section B.2.

### A.6 Choice of $L$

The choice of $L$ determines the extent of temporal coherence where $2L$ equals the input sequence length, and $L = 0$ corresponds to single inputs. For Table 1, we experimented with values of $L$ in the set $\{0, \frac{5}{36}S, \frac{1}{4}S, \frac{1}{2}S\}$ for both the TVAE and BubbleVAE. Both the BubbleVAE and TVAE achieved highest likelihoods at $L = \frac{5}{36}S$, and TVAE achieved lowest equivariance error at $L = \frac{1}{2}S$. We additionally included TVAE experiments with $L = \frac{13}{36}S$ for purposes of visualization in Figures 1 and 4 as this yielded the best qualitative generalization. For Table 2, we experimented with values of $L$ in the set $\{0, \frac{1}{6}S, \frac{4}{15}S, \frac{1}{3}S, \frac{2}{5}S, \frac{1}{2}S\}$ for both TVAE and BubbleVAE, and presented a broad selection in the table. The results of all models are shown in Section B below.

### A.7 Hyperparameter Selection

Hyperparameters such as learning rate, batch size, number of capsules, capsule size, and ultimately model architecture were chosen to allow for quick training on limited resources and were not tuned significantly. Since it was conceptually simpler to have an equal number of capsule dimensions and sequence elements, this limited the number of capsules we could then train efficiently. In Section C.1 we explain how a model with fewer capsule dimensions than sequence elements could be constructed with an alternative Roll operator. Additionally, from preliminary experiments, we observe that models with a number of internal capsule dimensions different from the number of sequence elements achieve similar likelihood values while also learning coherent transformations as decoded through the capsule roll. We believe these findings in combination with the extra studies provided in Section B suggest a satisfying degree of robustness to hyperparameter selection.

### A.8 MNIST Transformations

The first set of experiments presented in this paper are based on the MNIST dataset [37] (MIT Licence). For Section 6.2 (Figure 3) an MNIST training set of 48,000 images was used, while the standard test set of 10,000 images was used to compute the maximum activating image. For Section 6.3 (Figure 4 and Table 1), sequences of MNIST images were created by picking a random training image (with a random transformation 'pose') and successively transforming it according to one of the 3 available transformations (e.g. only one attribute is changed per sequence). The available transformations consisted of rotation, color (hue rotation), and scale with increments of 20-degrees for rotation and color, and $3.66\%$ increments for scale. Since scale is inherently non-cyclic, the bounds of the transformation were set at $60\%$ and $126\%$, and the transformations were constructed to be periodic such then once scale reached $126\%$, the next element was at $60\%$ scale. The final sequences were thus constructed to be 18 images long, where each element in the batch had an independently randomly chosen transformation. Again, the likelihood $\log p(\mathbf{x})$ and equivariance error $\mathcal{E}_{eq}$ were computed on the held-out 10,000 example test set, where the same random transformation sequences were applied.

### A.9 dSprites Transformations

The second set of experiments presented in this paper are based on the dSprites dataset [43] (Apache-2.0 License). To reduce computational complexity of this dataset, we took a subset of the dataset which consisted of all 3 shapes, the largest 5 scales, and every other example from the first 30 orientations, x-positions, and y-positions. The resulting dataset thus had 50,625 total images (3 shapes, 5 scales, 15 orientations, 15 x-positions, 15 y-positions), compared to the original 737,280 images. To construct sequences, we followed the same procedure as for MNIST, whereby first a random example and transformation were chosen, and a sequence of 15 images was constructed where only the chosen transformation was applied successively. We define the transformations available for sequences as scale, orientation, x-position, and y-position, omitting shape since smooth shape transforms are not present in the dSprites dataset. Again, we define all transformations to be cyclic such that once the 15th element is reached, the 1st element follows. For scale transformations, we simply loop over all 5 scales 3 times per sequence. We observe that although these sequences do not match the latent priors exactly, the models still train relatively well, implying some degree of robustness.

## A.10 Capsule Correlation Metric ($\mathrm{CapCorr}$)

Here we define $\mathrm{CapCorr}$ more precisely as it is implemented in our work. First, we denote the ground truth transformation parameter of the sequence at timestep $l$ as $y_l$ (e.g. the rotation angle at timestep $l$ for a rotation sequence), and the corresponding activation at time $l$ as $\mathbf{t}_l$. Next, to get an arbitrary starting point, we let $l = \Omega$ denote the timestep when $y_l$ is at its canonical position (e.g. rotation angle 0, x-position 0, or scale 1). We see $\Omega$ is not necessarily 0 since the first timestep of each sequence ($l = 0$) is a randomly transformed example. Then, we observe that we can measure the approximate observed roll in the capsule dimension between time 0 and $\Omega$ as a 'phase shift' by computing the index of the maximum value of a discrete (periodic) cross-correlation of $\mathbf{t}_\Omega$ and $\mathbf{t}_0$:

$$\mathrm{ObservedRoll}(\mathbf{t}_\Omega, \mathbf{t}_0) = \mathrm{argmax}\left[\mathbf{t}_\Omega \star \mathbf{t}_0\right] \tag{15}$$

Where $\star$ is discrete (periodic) cross-correlation across the (cyclic) capsule dimension and $\mathrm{argmax}$ is also subsequently performed over the capsule dimension. Then, the $\mathrm{CapCorr}$ metric for a single capsule is given as:

$$\mathrm{CapCorr}(\mathbf{t}_\Omega, \mathbf{t}_0, y_\Omega, y_0) = \mathrm{Corr}\left(\mathrm{ObservedRoll}(\mathbf{t}_\Omega, \mathbf{t}_0), |y_\Omega - y_0|\right) \tag{16}$$

Where the correlation coefficient $\mathrm{Corr}$ is then computed across all examples for the entire dataset. In our experiments we use the Pearson correlation coefficient for $\mathrm{Corr}$. We thus see this metric is the correlation of the estimated observed capsule roll with the shift in ground truth generative factors, which is equal to 1 when the model is perfectly equivariant. To extend this definition to multiple capsules, we estimate $\mathrm{ObservedRoll}$ for each capsule separately, and then correlate the mode of all $\mathrm{ObservedRoll}$ values with the true shift in ground truth generative factors. We see empirically that the $\mathrm{ObservedRolls}$ for all capsules are almost always identical (i.e. all capsules roll simultaneously for each transformation), therefore computing the mode does not destroy significant information. Finally, for transformation sequences which have multiple timesteps where $y_l$ is at the canonical position (e.g. scale transformations on dSprites where scale is looped 3 times), we select $l = \Omega$ to be the one from this possible set which yields the minimal absolute distance between $|y_\Omega - y_0|$ and $\mathrm{ObservedRoll}(\mathbf{t}_\Omega, \mathbf{t}_0)$.

## A.11 Definition of $\mathrm{Roll}$ for Capsules

As stated in Section 4.5.2, $\mathrm{Roll}_\delta(\mathbf{u})$, is defined as a cyclic permutation of $\delta$ steps along the capsule dimension of $\mathbf{u}$. Explicitly, if $\mathbf{u}$ is divided into $C$ capsules each with $D$ dimensions, the $\mathrm{Roll}_\delta$ operation can be written as:

$$\begin{aligned}
\mathrm{Roll}_\delta(\mathbf{u}) &= \mathrm{Roll}_\delta\left([u_1, u_2, \ldots, u_{C \cdot D}]\right) \\
&= [u_D, u_1, \ldots, u_{D-1}, u_{2 \cdot D}, u_{D+1}, \ldots, u_{2 \cdot D-1}, u_{3 \cdot D}, \ldots, \ldots u_{C \cdot D-1}]
\end{aligned} \tag{17}$$

# B Extended Results

In this section we provide extended results for all tested hyperparamters (Tables 3 & 4), a further analysis of the impact of the coherence window within a capsule $\mathbf{W}_\delta$ (Table 5), samples from the model in Section 6.2, and additional capsule traversal experiments highlighting the generalization capabilities of the TVAE to combinations of transformations unseen during training (Figure 6).

## B.1 Extended Tables 1 & 2

In Tables 3 & 4 below, we present extended versions of Tables 1 & 2 respectively, showing all tested settings of the TVAE & BubbleVAE. We observe the TVAE achieves perfect correlation ($\mathrm{CapCorr} = 1$) for $L \geq \frac{1}{3}$, and steadily decreasing correlation for lower values of $L$.

## B.2 Impact of $\mathbf{W}_\delta$

In Table 5, we show a small set of experiments with different settings of $\mathbf{W}_\delta$, and specifically changing values of $K$ (the coherence window within a capsule). As can be seen, increasing $K$ generally reduces equivariance error, but decreases the log-likelihood. This can be further understood by examining the capsule traversals of such models in Figures 9, 10, 11, 12, & 13. We see that larger values of $K$ appear to induce smoother transformations within the capsule dimensions, eventually resulting in invariant representations when $K$ is equal to the capsule dimensionality.

Table 3: Log Likelihood and Equivariance Error on MNIST for all models tested. Mean $\pm$ std. over 3 random initalizations.

| Model | TVAE | TVAE | TVAE | TVAE | TVAE |
|---|---|---|---|---|---|
| $L$ | $L = \frac{1}{2}S$ | $L = \frac{13}{36}S$ | $L = \frac{1}{4}S$ | $L = \frac{5}{36}S$ | $L = 0$ |
| $K$ | $K = 3$ | $K = 3$ | $K = 3$ | $K = 3$ | $K = 9$ |
| $\log p(\mathbf{x}) \uparrow$ | $\mathbf{-186.8} \pm 0.1$ | $-188.0 \pm 0.5$ | $-187.0 \pm 0.2$ | $\mathbf{-186.0} \pm 0.7$ | $-218.5 \pm 0.9$ |
| $\mathcal{E}_{eq} \downarrow$ | $\mathbf{573.9} \pm 1.5$ | $1089.8 \pm 2.4$ | $2136.9 \pm 7.8$ | $3246.6 \pm 3.3$ | $3216.6 \pm 104.9$ |

| Model | BubbleVAE | BubbleVAE | BubbleVAE | BubbleVAE | VAE |
|---|---|---|---|---|---|
| $L$ | $L = \frac{1}{2}S$ | $L = \frac{1}{4}S$ | $L = \frac{5}{36}S$ | $L = \frac{5}{36}S$ | $L = 0$ |
| $K$ | $K = 2L$ | $K = 2L$ | $K = 2L$ | $K = 3$ | $K = 1$ |
| $\log p(\mathbf{x}) \uparrow$ | $-200.9 \pm 0.7$ | $-202.3 \pm 1.4$ | $-190.8 \pm 0.7$ | $-191.4 \pm 0.5$ | $-189.0 \pm 0.8$ |
| $\mathcal{E}_{eq} \downarrow$ | $4206.7 \pm 903.3$ | $1141.7 \pm 9.6$ | $2605.7 \pm 16.1$ | $3369.5 \pm 11.9$ | $13273.9 \pm 0.5$ |

Table 4: Equivariance error and $\mathrm{CapCorr}$ for all models tested on the dSprites dataset. Mean $\pm$ standard deviation over 3 random initalizations.

| Model | TVAE | TVAE | TVAE | TVAE | TVAE | TVAE |
|---|---|---|---|---|---|---|
| $L$ | $L = \frac{1}{2}S$ | $L = \frac{2}{5}S$ | $L = \frac{1}{3}S$ | $L = \frac{4}{15}S$ | $L = \frac{1}{6}S$ | $L = 0$ |
| $K$ | $K = 1$ | $K = 1$ | $K = 1$ | $K = 1$ | $K = 1$ | $K = 3$ |
| $\mathrm{CapCorr}_X \uparrow$ | $\mathbf{1.0} \pm 0$ | $\mathbf{1.0} \pm 0$ | $\mathbf{1.0} \pm 0$ | $0.95 \pm 0.00$ | $0.67 \pm 0.02$ | $0.17 \pm 0.03$ |
| $\mathrm{CapCorr}_Y \uparrow$ | $\mathbf{1.0} \pm 0$ | $\mathbf{1.0} \pm 0$ | $\mathbf{1.0} \pm 0$ | $0.96 \pm 0.01$ | $0.66 \pm 0.02$ | $0.21 \pm 0.02$ |
| $\mathrm{CapCorr}_O \uparrow$ | $\mathbf{1.0} \pm 0$ | $\mathbf{1.0} \pm 0$ | $\mathbf{1.0} \pm 0$ | $0.88 \pm 0.01$ | $0.52 \pm 0.01$ | $0.09 \pm 0.01$ |
| $\mathrm{CapCorr}_S \uparrow$ | $\mathbf{1.0} \pm 0$ | $\mathbf{1.0} \pm 0$ | $\mathbf{1.0} \pm 0$ | $0.96 \pm 0.01$ | $0.42 \pm 0.01$ | $0.51 \pm 0.01$ |
| $\mathcal{E}_{eq} \downarrow$ | $\mathbf{344} \pm 5$ | $759 \pm 9$ | $1034 \pm 6$ | $1395 \pm 7$ | $2549 \pm 38$ | $2971 \pm 9$ |

| Model | BubbleVAE | BubbleVAE | BubbleVAE | BubbleVAE | BubbleVAE | VAE |
|---|---|---|---|---|---|---|
| $L$ | $L = \frac{1}{2}S$ | $L = \frac{2}{5}S$ | $L = \frac{1}{3}S$ | $L = \frac{4}{15}S$ | $L = \frac{1}{6}S$ | $L = 0$ |
| $K$ | $K = 2L$ | $K = 2L$ | $K = 2L$ | $K = 2L$ | $K = 2L$ | $K = 1$ |
| $\mathrm{CapCorr}_X \uparrow$ | $0.16 \pm 0.01$ | $0.15 \pm 0.01$ | $0.13 \pm 0.01$ | $0.12 \pm 0.02$ | $0.09 \pm 0.01$ | $0.18 \pm 0.01$ |
| $\mathrm{CapCorr}_Y \uparrow$ | $0.15 \pm 0.01$ | $0.14 \pm 0.01$ | $0.12 \pm 0.01$ | $0.12 \pm 0.01$ | $0.11 \pm 0.02$ | $0.16 \pm 0.01$ |
| $\mathrm{CapCorr}_O \uparrow$ | $0.12 \pm 0.00$ | $0.13 \pm 0.02$ | $0.10 \pm 0.01$ | $0.09 \pm 0.00$ | $0.06 \pm 0.01$ | $0.11 \pm 0.00$ |
| $\mathrm{CapCorr}_S \uparrow$ | $0.52 \pm 0.02$ | $0.55 \pm 0.00$ | $0.52 \pm 0.00$ | $0.48 \pm 0.02$ | $0.27 \pm 0.01$ | $0.52 \pm 0.00$ |
| $\mathcal{E}_{eq} \downarrow$ | $6825 \pm 126$ | $6917 \pm 13$ | $1951 \pm 34$ | $2181 \pm 627$ | $1721 \pm 27$ | $6934 \pm 0$ |

Table 5: Impact of $\mathbf{W}_\delta$ (i.e. $K$) on MNIST performance.

| Model | TVAE | TVAE | TVAE | TVAE | TVAE |
|---|---|---|---|---|---|
| $L$ | $L = \frac{5}{36}S$ | $L = \frac{5}{36}S$ | $L = 0$ | $L = 0$ | $L = 0$ |
| $K$ | $K = 3$ | $K = 9$ | $K = 3$ | $K = 9$ | $K = 18$ |
| $\log p(\mathbf{x}) \uparrow$ | $\mathbf{-186.0} \pm 0.7$ | $-190.6 \pm 0.2$ | $-213.4 \pm 1.2$ | $-218.5 \pm 0.9$ | $-224.8 \pm 1.0$ |
| $\mathcal{E}_{eq} \downarrow$ | $3246.6 \pm 3.3$ | $2606.3 \pm 17.0$ | $12085.7 \pm 68.5$ | $3216.6 \pm 104.9$ | $1090.3 \pm 19.3$ |

### B.3 Samples

In Figure 5, we provide samples from our model in the $L = 0$ setting to validate that the learned latent distribution closely matches the $TPoT$ distribution described in Equation 6. Explicitly, the samples are generated by sampling standard normal random variables $\mathbf{Z}$ and $\mathbf{U}$, constructing $\mathbf{T}$ as in Equation 6, and then passing these sampled $\mathbf{T}$ through the decoder. We see that the samples resemble true MNIST digits (accounting for the limited capacity of the model), implying that the distribution after training indeed follows the desired distribution, and the model has learned to become a good generative model of the data.

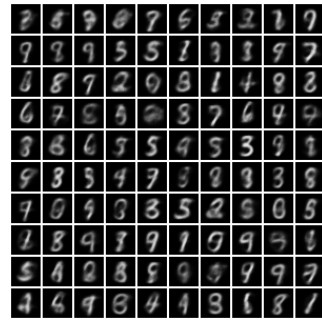

Figure 5: Samples from the TVAE in Section 6.2.

### B.4 Generalization to Combined Transformations at Test Time

In this section, we test the ability of the model to generate sequences composed of multiple transformations through a capsule roll, despite only being trained on individual transformations in isolation. In other words, we intend to measure the extent to which the transformations learned by a set of capsules can be combined simply by passing input sequences with corresponding combined transformations. Such generalization suggests powerful benefits to data efficiency, effectively factorizing a set of complex transformations.

Explicitly, we train the model identically to that presented in Figure 4, (TVAE $L = \frac{13}{36}S$), and examine the sequences generated by a capsule roll when the partial input sequences contain combinations of transformations previously unseen during training. The results of this experiment, tested on combinations of rotation and color transforms on the MNIST test set, are presented in Figure 6 below. Although this generalization capability is not known to be guaranteed a priori, we see that the capsule traversals are frequently remarkably coherent with the input transformation, implying that the model may indeed be able to generalize to combinations of transformations. Furthermore, we observe with $L = \frac{1}{2}S$ (results not shown), this generalization capability is nearly perfect.

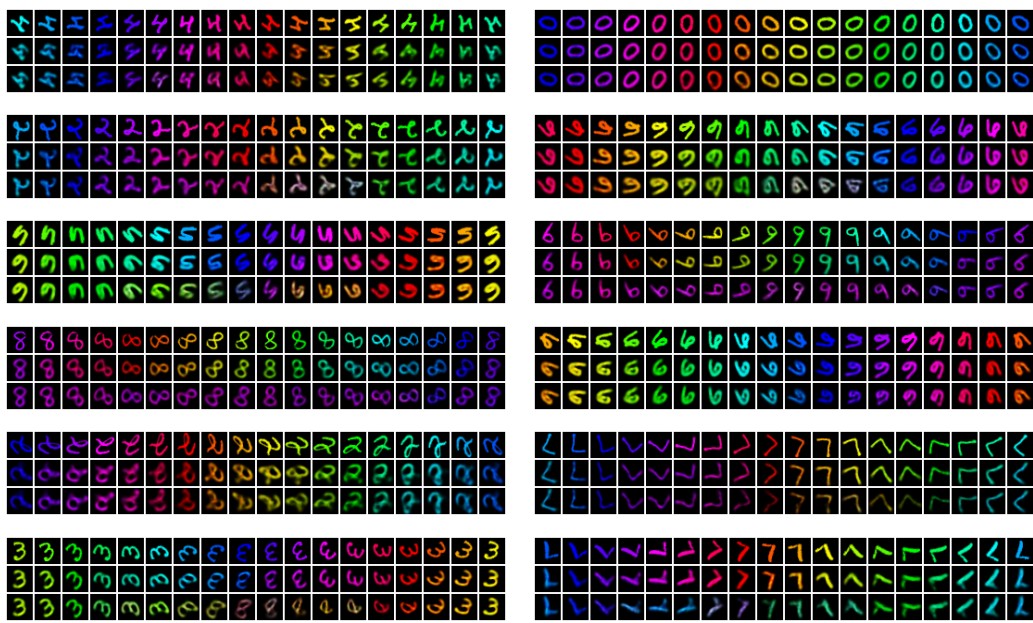

Figure 6: Capsule Traversals for MNIST TVAE $L = \frac{13}{36}S$, trained on individual transformations in isolation, and tested on combined color and rotation transformations. Top row shows the input sequence, middle row shows the direct reconstruction $\{g_\theta(\mathbf{t}_l)\}_l$, and bottom row shows the capsule traversal $\{g_\theta(\mathrm{Roll}_l[\mathbf{t}_0])\}_l$.

## C  Proposed Model Extensions

### C.1  Extensions to Roll & CapCorr

The Roll operation can be seen as defining the speed at which $\mathbf{t}$ transforms corresponding to an observed transformation. For example, with Roll defined as in Section A.11 above, we implicitly assume that for each observed timestep, we would like the representation $\mathbf{t}$ to cyclically permute 1-unit within the capsule. For this to match the observed data, it requires the model to have an equal number of capsule dimensions and sequence elements. If we wish to reduce the size of our representation, we could instead encourage a 'partial permutation' for each observed transformation. For a single capsule with $D$ elements, an example of a simple linear version of such a partial permutation (for $0 < \alpha \leq 1$) can be implemented as:

$$\text{Roll}_\alpha(\mathbf{u}) = \big[\alpha u_D + (1-\alpha)u_1, \ \alpha u_1 + (1-\alpha)u_2, \ \ldots, \ \alpha u_{D-1} + (1-\alpha)u_D\big] \qquad (18)$$

A slightly more principled partial roll for periodic signals could also be achieved by performing a phase shift of the signal in Fourier space, and performing the inverse Fourier transform to obtain the resulting rolled signal. To extend the CapCorr metric to similarly allow for partial Rolls, we see that we can simply redefine the ObservedRoll (originally given by discrete cross-correlation) to be given by the argmax of the inner product of a sequentially partially rolled activation with the initial activation $\mathbf{t}_\Omega$. Formally:

$$\text{ObservedRoll}(\mathbf{t}_\Omega, \mathbf{t}_0) = \text{argmax}\,[\mathbf{t}_\Omega \cdot \text{Roll}_0(\mathbf{t}_0), \ \mathbf{t}_\Omega \cdot \text{Roll}_\alpha(\mathbf{t}_0), \ \ldots, \mathbf{t}_\Omega \cdot \text{Roll}_{D-\alpha}(\mathbf{t}_0)] \quad (19)$$

### C.2  Non-Cyclic Capsules

We can also see that there is nothing beyond convenience which inherently requires the capsules to be circular (i.e. have periodic boundary conditions). To implement linear capsules, we propose one solution is to add $L$ additional $U_i$ variables to both the left and right boundaries of each capsule. In this way, the vector $\mathbf{U}$ is larger than the vector $\mathbf{Z}$ and can be seen as a 'padded' version, where the padding is composed of independant random variables. Additionally, the transformation sequences can then be padded on both sides by replicating the first and final elements $L$ times. The construction of $\mathbf{T}$ variables is then performed identically as in Equations 7 and 9. The Roll operation can then be similarly defined as filling the boundaries with 0 since these values will not be used as part of the computation.

### C.3  Multi-dimensional Temporally Coherent Capsules

In consideration of transformations which may naturally live in multiple dimensions, we wish to extend the original model to support multi-dimensional capsules. Such multi-dimensional capsules could additionally support more well-defined 'disentanglement' of transformations by encouraging each transformation to be axis-aligned with one dimension of each capsule. We see that in the non-temporally coherent case ($L = 0$), the model can easily be extended to capsules of multiple dimensions through multi-dimensional neighborhoods. An example of a model with 2-dimensional neighborhoods is presented in Figure 3. However, when considering shifting temporal coherence as we defined in Section 6.3, it is not clear how the shift operator or the neighborhoods should be defined for higher dimensional capsules. In this section we propose to modify the definitions of $\mathbf{T}$ in Equations 7 and 9 with an extension resembling 'group sparsity' in the denominator.

First, we again assume that each input sequence is an observation of a single transformation at a time. Formally, the multi-dimensional capsules are then constructed by arranging $\mathbf{U}$ into a $D$ dimensional lattice. In such a model, we desire to roll and sum only along a single axis of the lattice for a given sequence. Incorporating this into the construction of $\mathbf{T}$ yields the following:

$$\mathbf{T}_l = \frac{\mathbf{Z}_l - \mu}{\sum_{d=1}^{D} \sqrt{\mathbf{W}^d\left[\mathbf{U}_{l+L}^2; \cdots ; \mathbf{U}_{l-L}^2\right]}} = \frac{\mathbf{Z}_l - \mu}{\sum_{d=1}^{D} \sqrt{\sum_{\delta=-L}^{L} \mathbf{W}_\delta^d \text{Roll}_\delta^d(\mathbf{U}_{l+\delta}^2)}} \qquad (20)$$

Where $\mathbf{W}_\delta^d$ refers to a matrix which sums locally along the $d^{th}$ dimension of each capsule, and not at all along the others, and similarly $\text{Roll}_\delta^d$ rolls only along the $d^{th}$ dimension. In practice we observe such models can indeed disentangle up to 2 distinct transformations, but become more challenging to optimize for higher dimensions. We believe this is potentially due to the exponential growth in capsule size with increasing dimension, but leave further exploration to future work.

## C.4 Causal Temporal Coherence

As noted in the limitations, the sequence model in this paper is not 'causal', meaning that each variable $\mathbf{T}_l$ requires variables from future timesteps in the sequence ($\mathbf{U}_{l+\delta}$ for $\delta > 0$). Although for the purpose of learning equivariance in practice this may not be an issue, it may be relevant for some online learning applications. We can modify Equations 7 and 9 by changing the matrix $\mathbf{W}$ (implemented as convolution) to a causal convolution (i.e. masking out $\mathbf{W}_\delta$ for $\delta > 0$). Formally:

$$\mathbf{T}_l = \frac{\mathbf{Z}_l - \mu}{\sqrt{\mathbf{W}\left[\mathbf{U}_l^2; \cdots; \mathbf{U}_{l-L}^2\right]}} = \frac{\mathbf{Z}_l - \mu}{\sqrt{\sum_{\delta=-L}^{0} \mathbf{W}_\delta \mathrm{Roll}_\delta(\mathbf{U}_{l+\delta}^2)}} \tag{21}$$

In a causal setting, it is also likely the transformations are no longer assumed to be circular. We thus refer the reader to Section C.2 above on non-circular capsules, which can be combined with Equation 21, to achieve such a model.

## D   Capsule Traversals

In this section we provide a set of 12 capsule traversals for each of the models presented in main text. The traversals are randomly selected such that all transformations (and dSprites shapes) are shown evenly. Unlike the main section, we additionally include a middle row which shows the direct reconstruction of the input without any rolling (i.e. $\{g_\theta(\mathbf{t}_l)\}_l$). We find the direct reconstructions valuable to determine if poor traversals are due to bad reconstructions (low $\log p_\theta(\mathbf{x}|\mathbf{t})$) or a lack of equivariance (high $\mathcal{E}_{eq}$). For example, with the baseline VAE models, we see that the reconstructions in the middle row are accurate for the full sequence, while the capsule traversals obtained by sequentially rolling the initial activation (shown in the bottom row) are nothing like the input transformation (top row). In all traversals, the left-most image corresponds to $\mathbf{t}_0$, and thus input sequences of length $2L$ cover both the left and right edges when $L > 0$.

Finally, in Figures 22 & 23 at the end of the section, we include capsule traversals for models trained on MNIST with more complex transformations such as combined color & rotation, and combined color & perspective transforms. These models were trained in an identical manner to the other MNIST models, with the same architecture, only changing the transformation sequences of the training dataset.

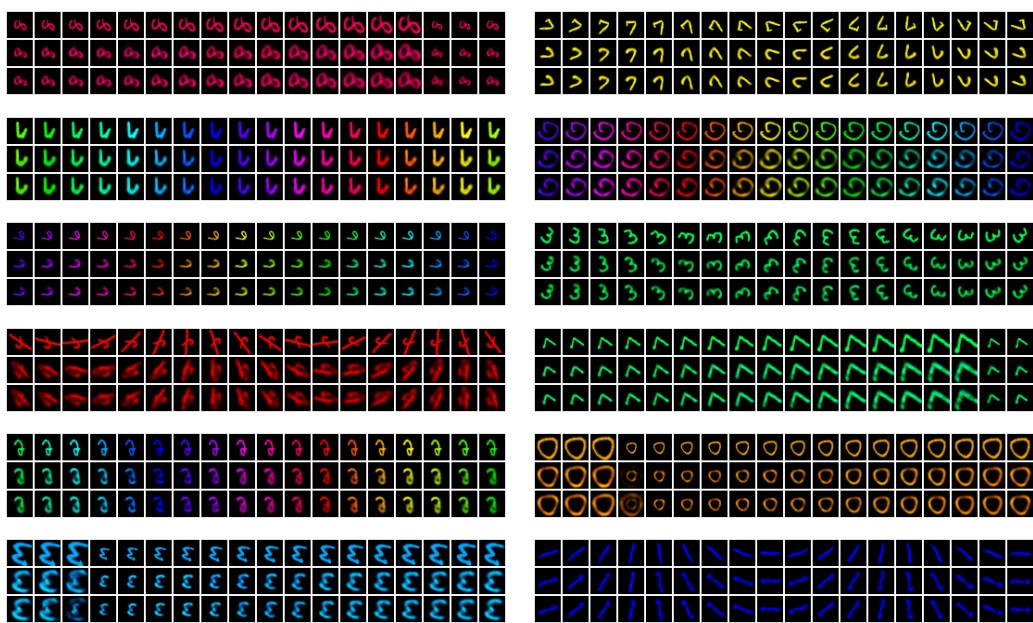

Figure 7: MNIST TVAE $L = \frac{1}{2}S$, $K = 3$

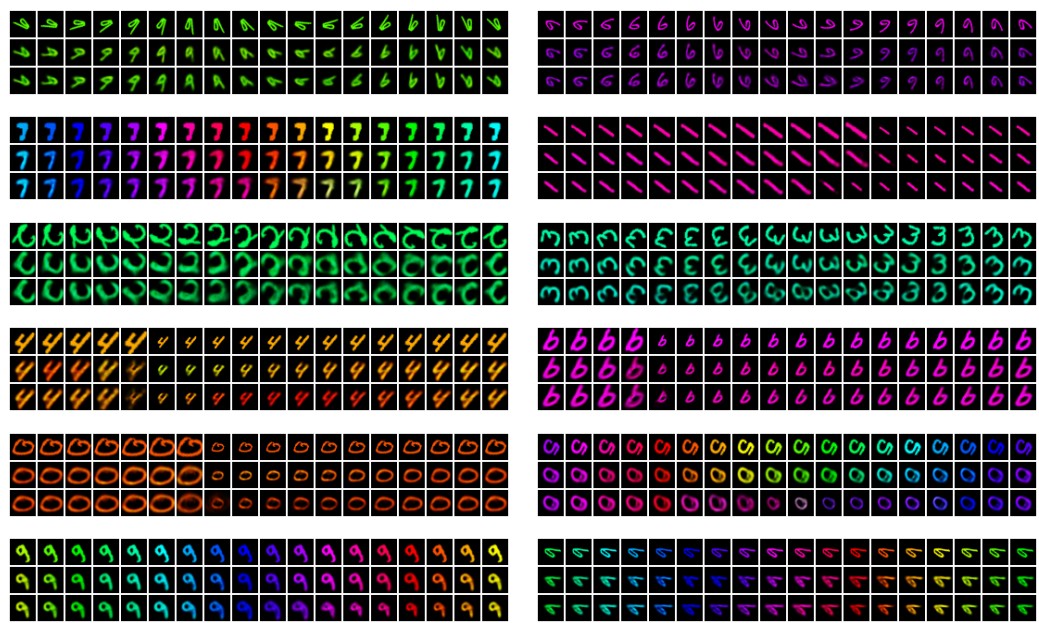

Figure 8: MNIST TVAE $L = \frac{13}{36}S$, $K = 3$

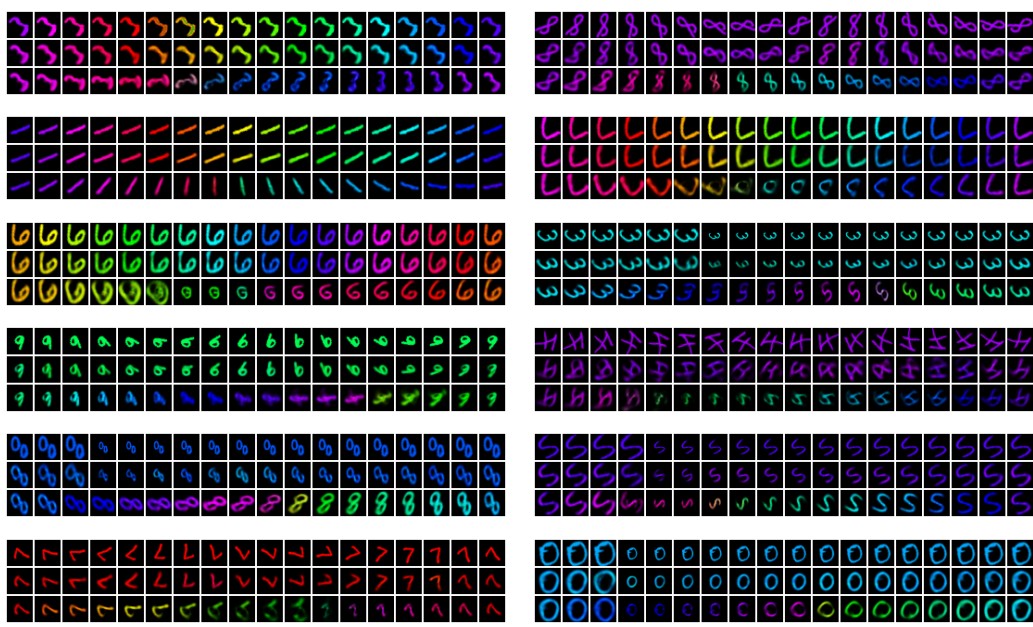

Figure 9: MNIST TVAE $L = \frac{5}{36}S$, $K = 3$. We see with values of $L < \frac{1}{3}S$ the transformations decoded through the capsule roll are only partially coherent with the input sequence.

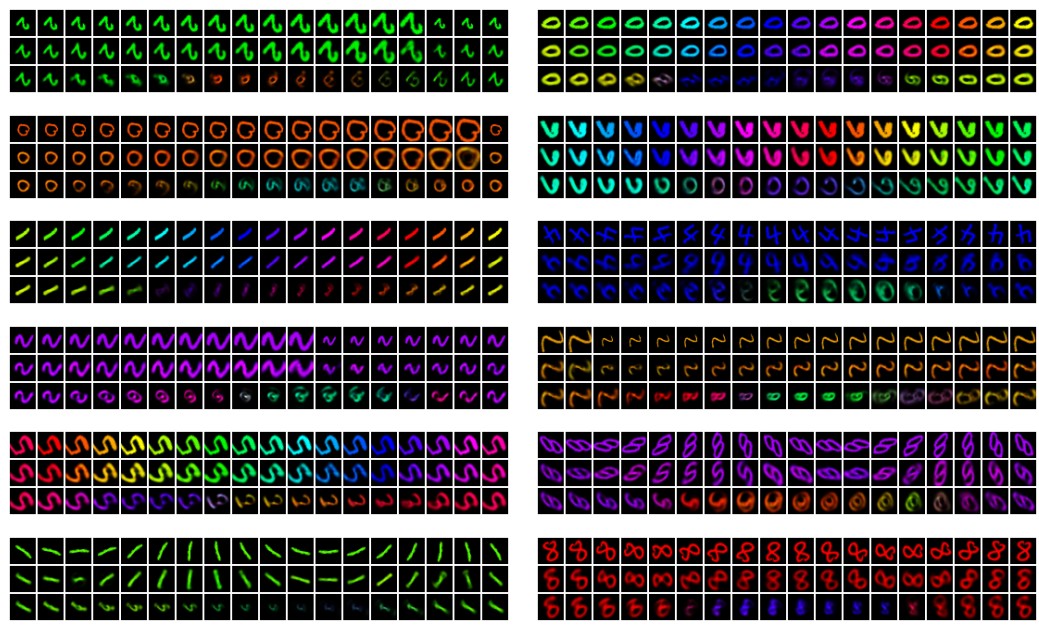

Figure 10: MNIST TVAE $L = \frac{5}{36}S$, $K = 9$

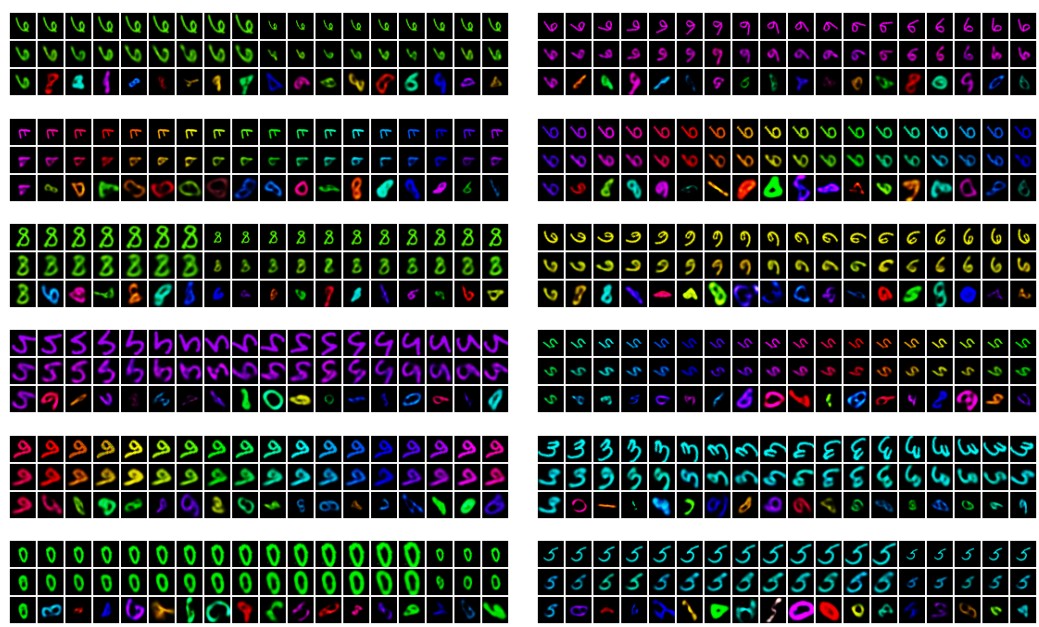

Figure 11: MNIST TVAE $L = 0$, $K = 3$. We see for sufficiently small values of $K$, the TVAE can reach a degenerate solution where topographic organization is almost entirely lost.

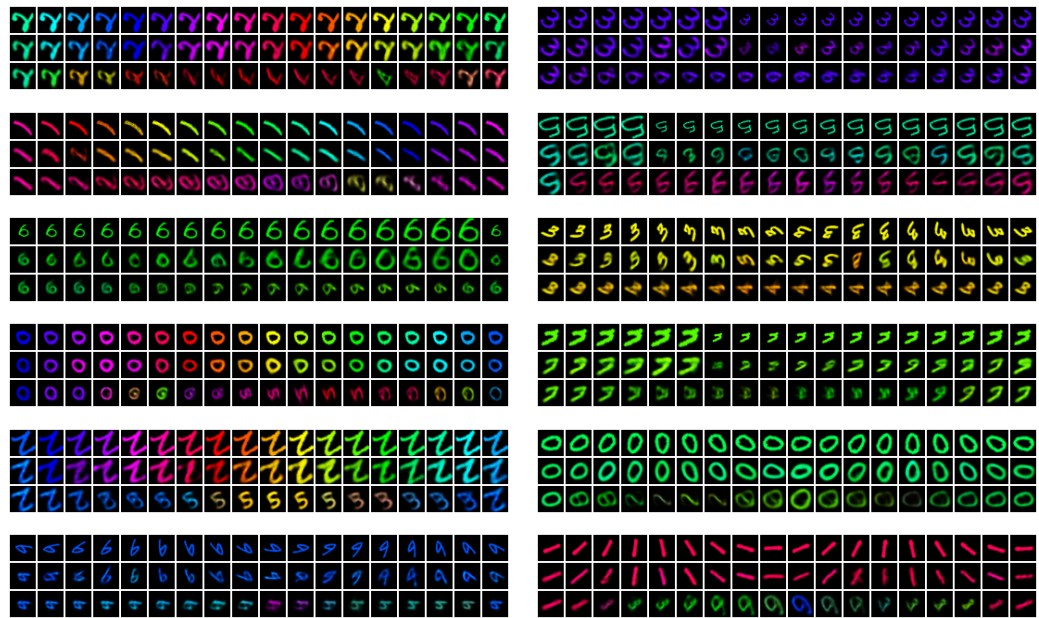

Figure 12: MNIST TVAE $L = 0$, $K = 9$

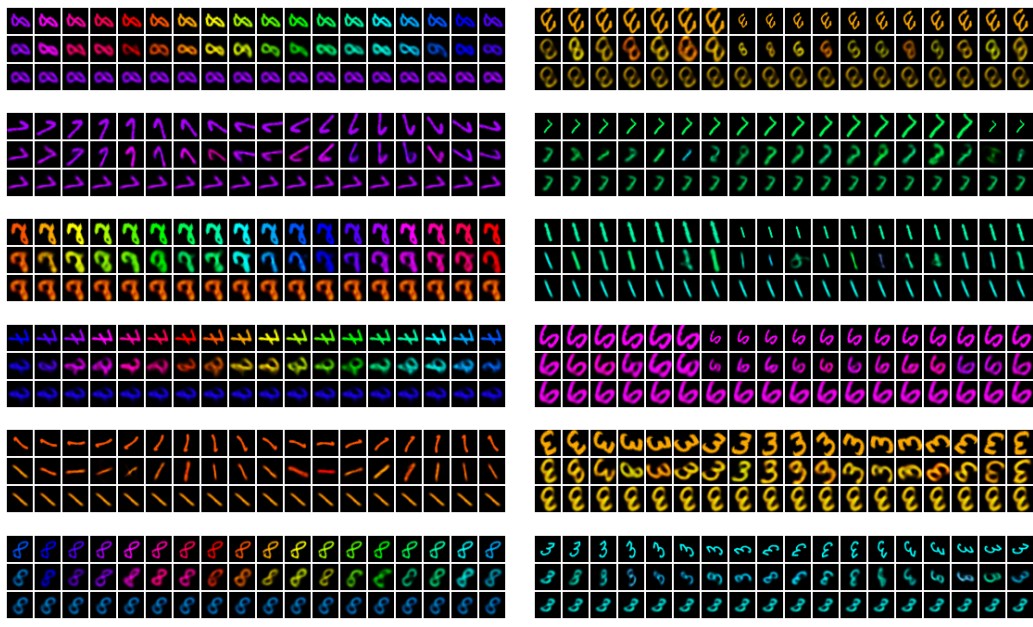

Figure 13: MNIST TVAE $L = 0$, $K = 18$. We see when $K$ is equal to the capsule size (making the model analogous to ISA), the model learns an invariant capsule representation – meaning Rolling a capsule activation produces no significant transformation in the observation space.

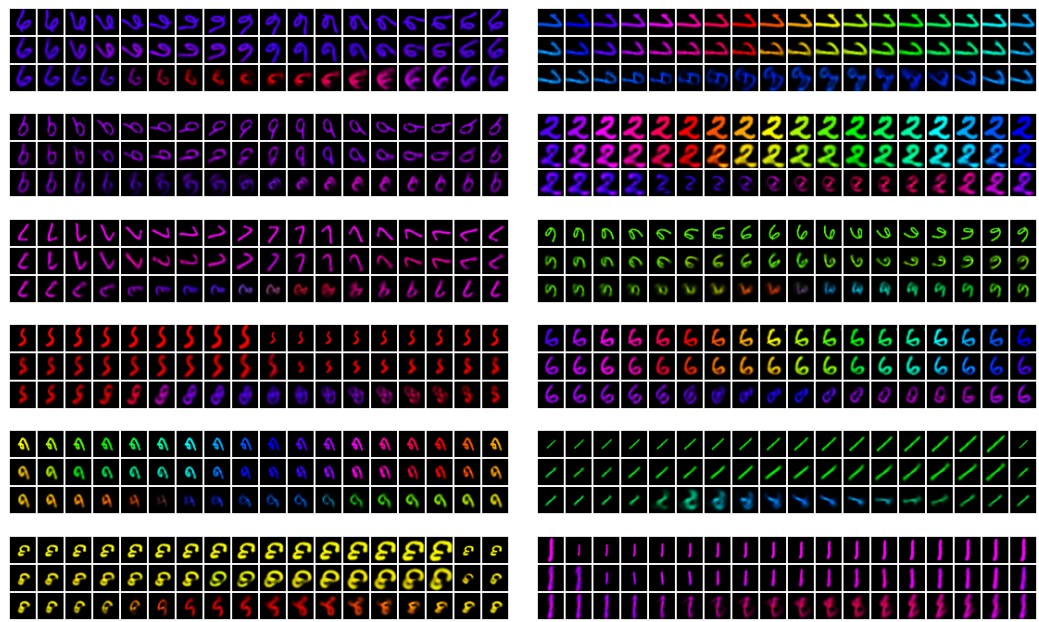

Figure 14: MNIST BubbleVAE $L = \frac{5}{36}S$, $K = 2L$

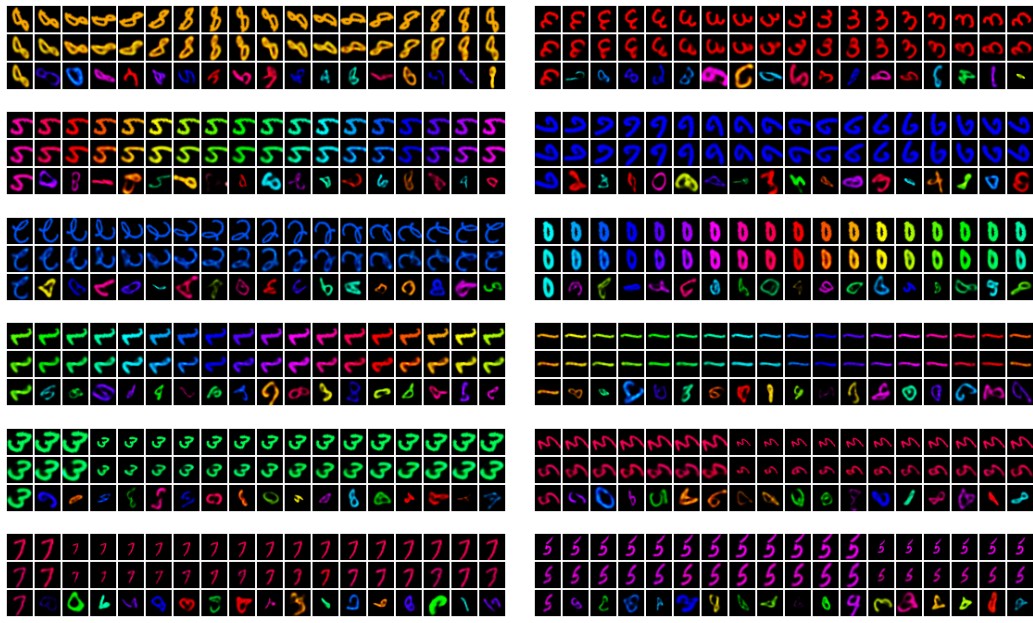

Figure 15: MNIST VAE $L = 0$, $K = 1$. We see images generated through capsule traversal with the baseline VAE appear entirely random, as expected due to the non-topographic nature of the VAE's latent space.

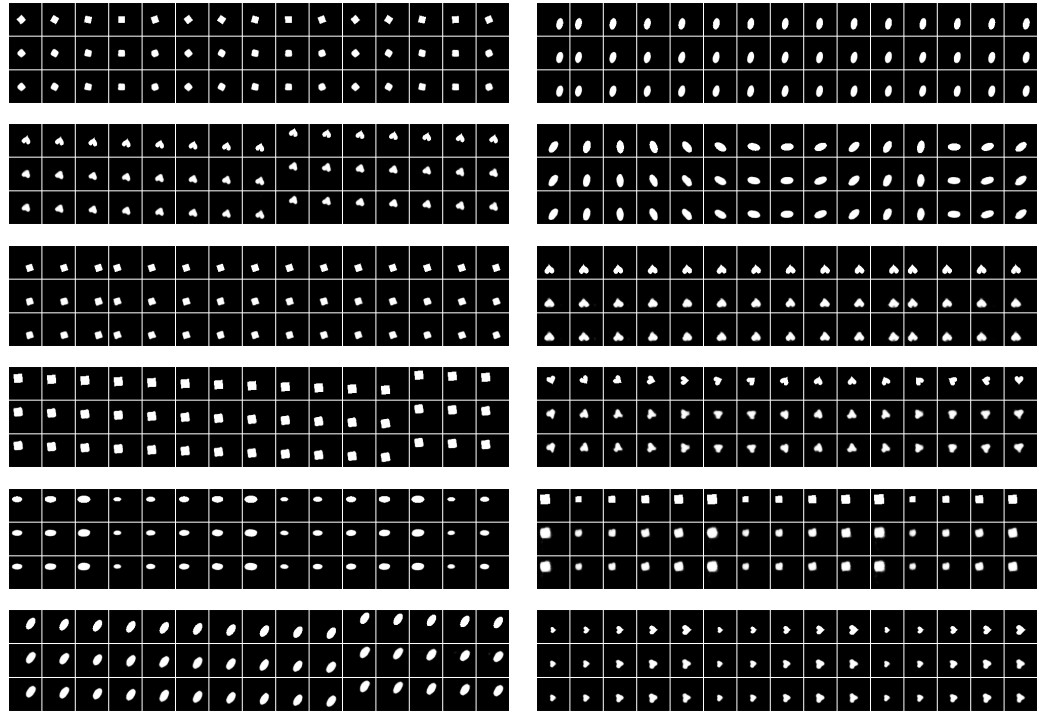

Figure 16: dSprites TVAE $L = \frac{1}{2}S$, $K = 1$

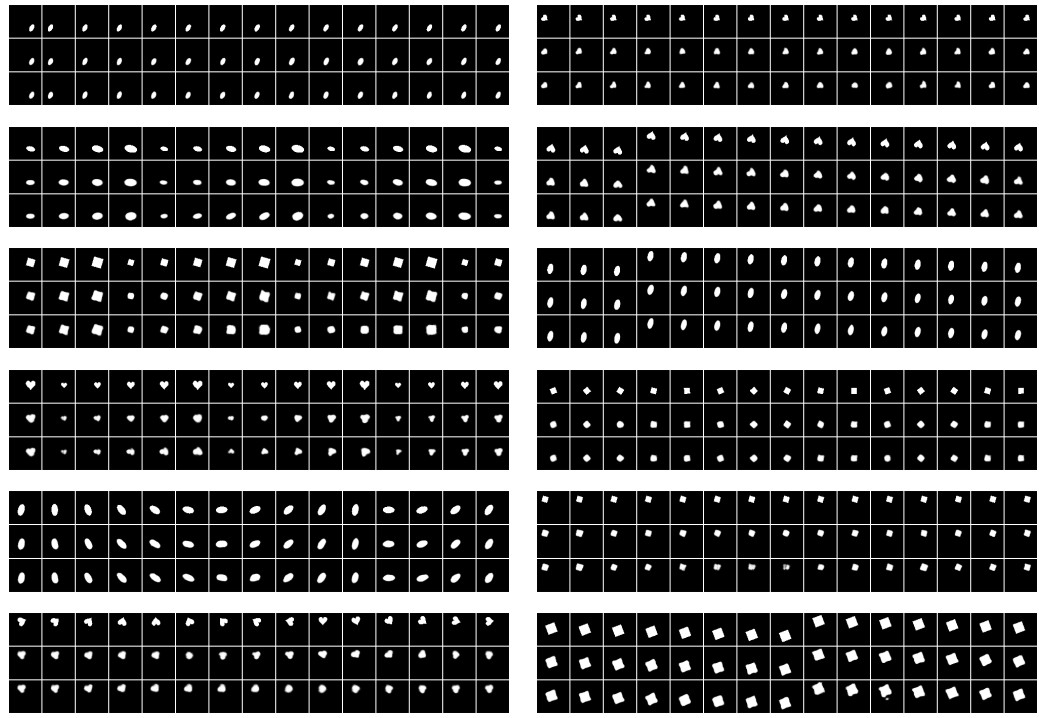

Figure 17: dSprites TVAE $L = \frac{1}{3}S$, $K = 1$

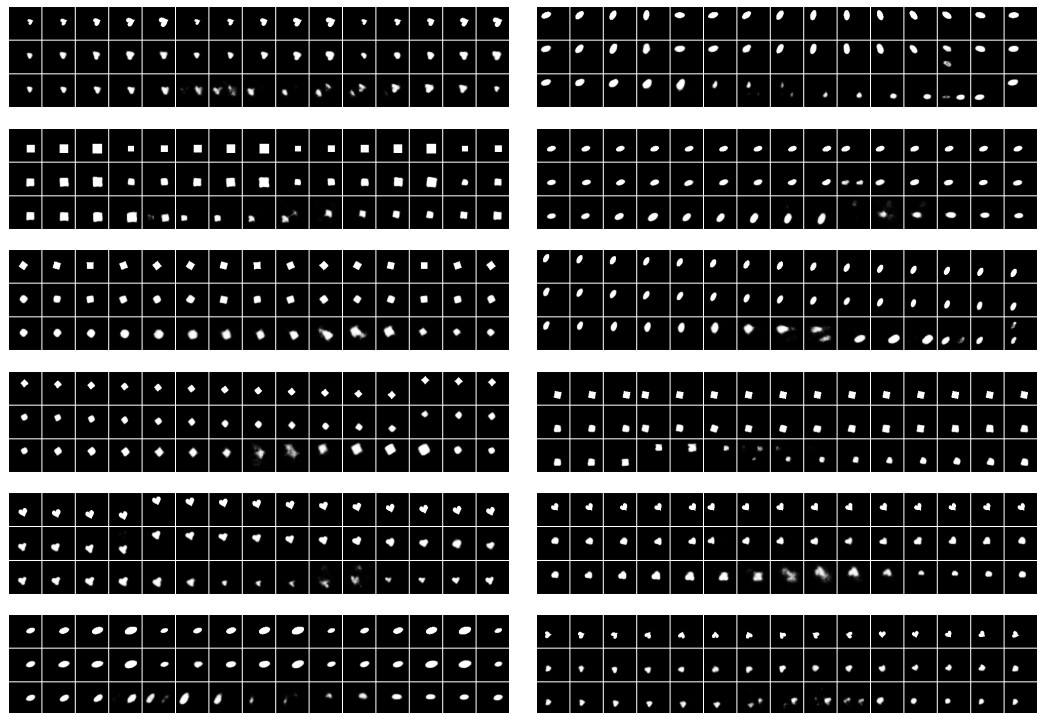

Figure 18: dSprites TVAE $L = \frac{1}{6}S$, $K = 1$

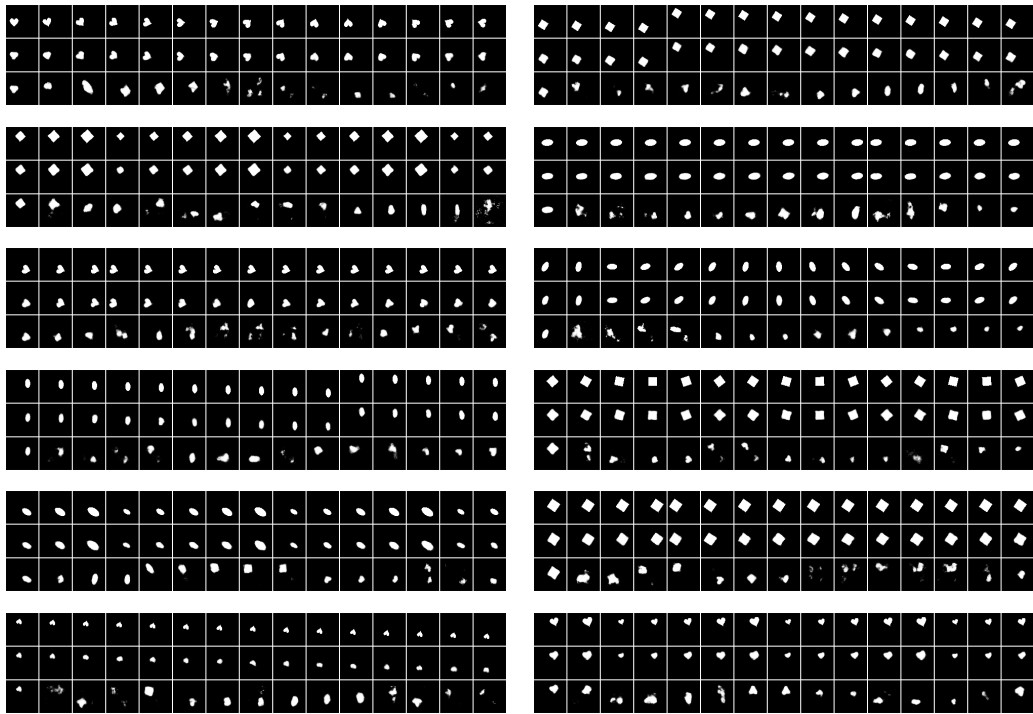

Figure 19: dSprites TVAE $L = 0$, $K = 3$

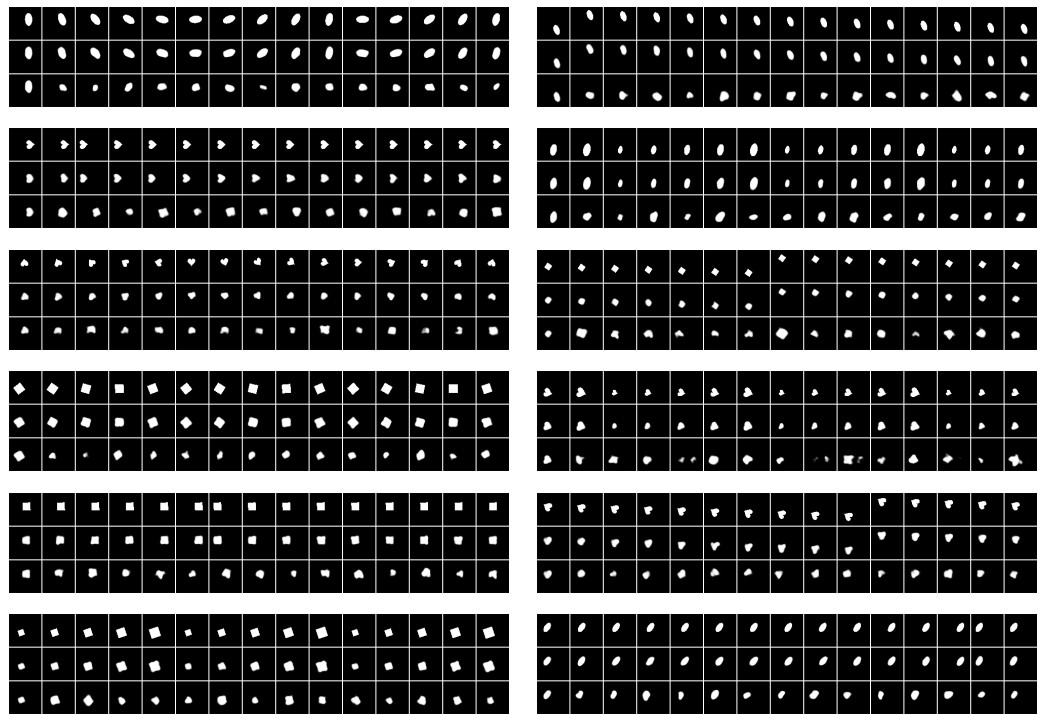

Figure 20: dSprites BubbleVAE $L = \frac{1}{3}S$, $K = 2L$. We see the capsule traversals for the BubbleVAE produce only relatively minor transformations in the observation space (e.g. shape or rotation change, but position appears constant). This reinforces the intuition that models with stationary temporal coherence are likely to learn invariant capsule representations.

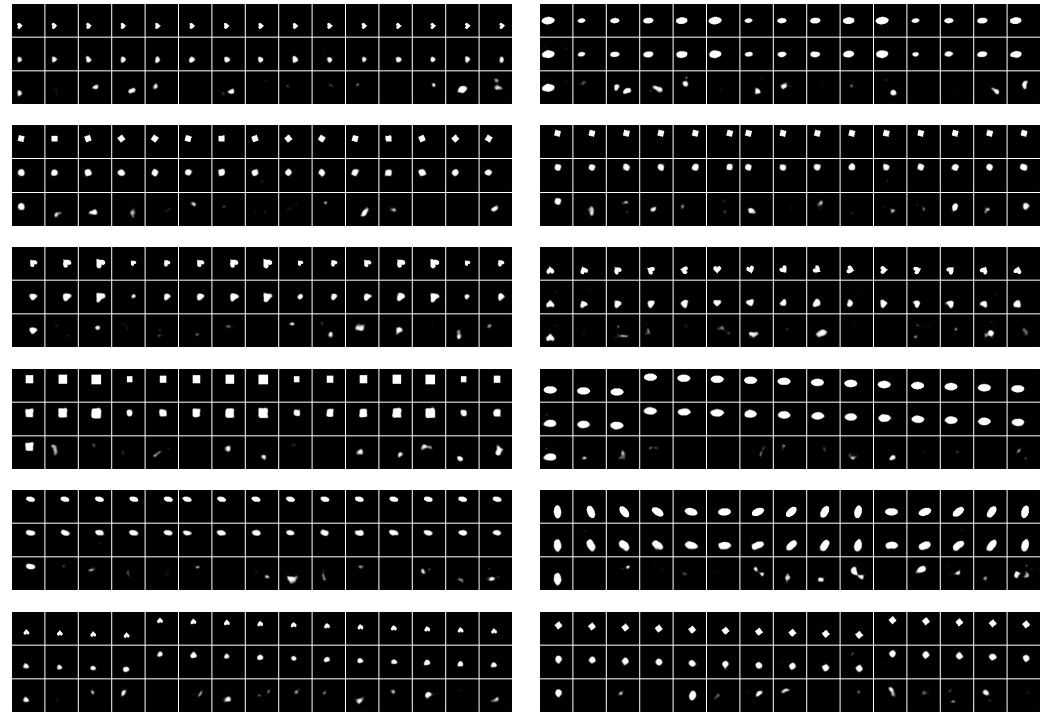

Figure 21: dSprites VAE $L = 0$, $K = 1$

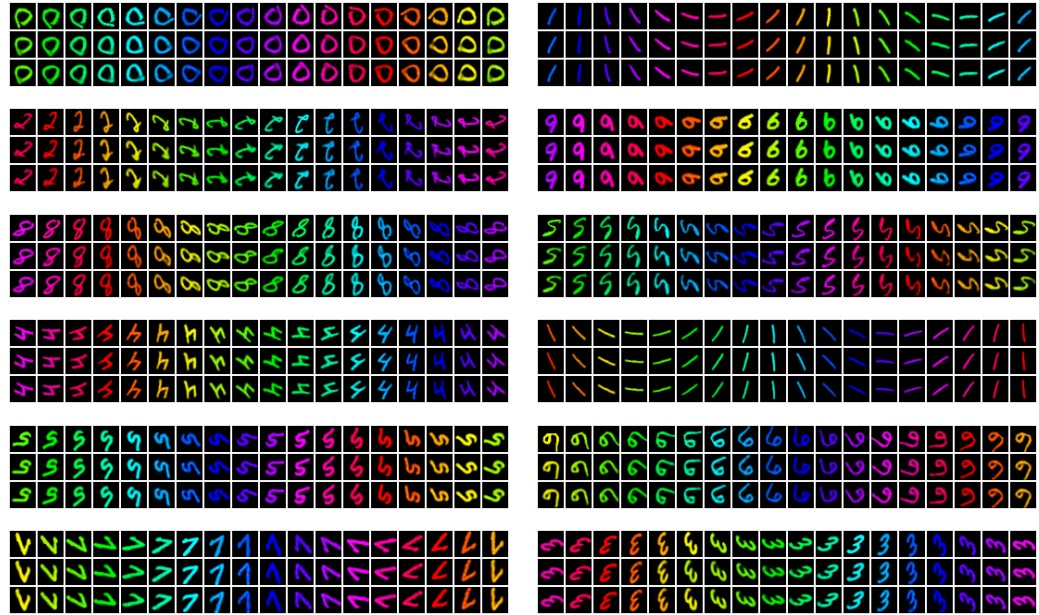

Figure 22: Combined Color & Rotation MNIST TVAE $L = \frac{13}{36}S$, $K = 3$. We see these generated sequences are slightly more accurate than those in Figure 6. This is to be expected since the model in this figure is trained explicitly on combinations of transformations, whereas the model in Figure 6 was trained on transformations in isolation, and tested on combinations to explore its generalization.

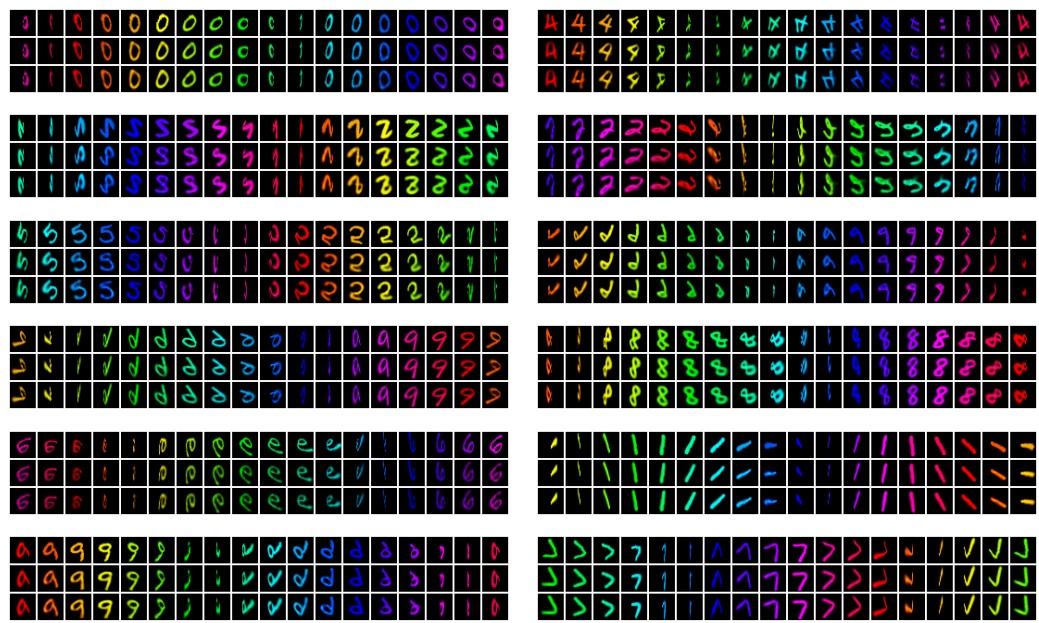

Figure 23: Combined Color & Perspective MNIST TVAE $L = \frac{13}{36}S$, $K = 3$. We see the TVAE is able to additionally learn combinations of complex transformations (like out-of-plane rotation) without any changes to the training procedure other than a change of dataset.