# OpenReview forum: "Topographic VAEs learn Equivariant Capsules"
_NeurIPS.cc/2021/Conference — NeurIPS 2021 Poster_

### Official Review · Reviewer_P3ig · 2021-07-16

**Rating:** 5
**Confidence:** 3

**Summary:**

In their work, the authors propose a novel VAE-based model named "Topographic VAE", which is inspired by topographic ICA and based on the Topographic Product of Student's t (TPoT) distribution. The authors show how topographic correlations following the TPoT distribution can be modelled by combining independent, normally distributed random variables across local neighbourhoods. This way, a model based on a latent TPoT distribution can be learned in a variational framework similar to standard VAEs. The authors highlight the similarity to capsules when considering multiple subspaces with independent local neighbourhood structures. Inspired by previous works on temporal coherence, the model is extended to image sequences by using spatiotemporal neighbourhood structures. When using a neighbourhood that is shifted for each time step, the authors show that transformations in the input sequences can be projected into the future by applying shifts to the encoded latent code. So the latent of the model is shown to be equivariant w.r.t. the transformations observed in the training data.

**Limitations And Societal Impact:**


- The authors discuss the limitations of the model honestly in Section 7, and the potential societal impact of the work in Section 9.
- As detailed above, the limitation and effects stemming from the fact that the model can only encode the latent $T$ for sequences should be adressed in the experimental section. The discussion in the limitations section is adequate.
- The discussion of the potential societal impact is brief, but given the rather theorectical nature of the paper I think it is okay.

**Main Review:**


### Originality
- The contribution of the work is the integration of topographic modelling with VAEs, and extending temporal coherence to learn equivariant representations. To my knowledge, this has not been done before.
- The idea of topographic models, also including temporal coherence, is not new. The authors thoroughly cite the previous literature on the topic, both in a dedicated related work section and when explaining the model components.

### Quality
- To me, the derivation of the model is well justified and related to prior work with great care. Moreover, the metrics proposed for evaluating the equivariance properties of the model are technically sound.
- The authors evaluate several variants of the model on two datasets (MNIST variant, dSprites) to evaluate the usefulness of the novel modelling components. I have however some concerns regarding the empirical evaluation of the model. In my view, some important questions regarding the representation learned by the model are not answered in the paper:
    - No samples from the model are shown or evaluated. The model is generative and VAE based, and the derivation of the TPoT distribution requires independent and normally distributed random variables. When training VAE based models in practice, fitting the factorized normal distribution and obtaining low reconstruction error frequently have to be traded against each other. In the paper, only the reconstruction capabilities of the model are evaluated. So there's no control whether the latent variables actually follow the intended distribution after training.
    - The aim of the shifting temporal coherence is to learn an equivariant representation of the training sequences. Only one of the transformations is changed in every training sequence, the shifting constraint however always affects all latent variables. To me it is not clear how the constant parameters are represented by the model, especially since the encoders are independent for every image according to equations (10) and (11). An anlysis of the representation in this respect would contribute to a better understanding the model.
    - As discussed with the limitations, the model is only able to infer the latent representation $T$ for image sequences. It would be interesting to know how the representation of the same image changes when it is part of different sequences. Moreover, the representation $T$ of a sequence could be sampled given $z$ and $u$ for a fixed image. This kind of experiments could also help to yield insights regarding the previous issue.
    - How does the model depend on the size of the latent space, i.e. the number and dimensionality of the capsules?
    - How is an image transformed, if the latent representation is rolled only for a single capsule? Does this correspond to meaningful transformations?


### Clarity
- Overall, the paper is very clearly written, well-structured and provides the necessary background for readers not familiar to topographic modelling.
- In my view, more details about the models used for the experiments, especially on the choice of neighbourhood structures and the weight normalization, should be provided in the main paper. To obey the page limit, the background and derivation of the model could be described more compactly.
- Figure 1: I do not understand the visualizion of the rolls in the latent space. Visualizing more latent variables per capsule, or smaller shifts would make the figure clearer.
- Some minor issues:
    - Figure 2: There should not be a line between the $Z$s as they are independent.
    - L217: Missing reference / undefined BibTeX entry
    - Equation 1 and 10: Whats the role of $g_\theta(t)$? Can it be subsumed into $p_\theta(x|t)$?

### Significance
- As stated by the authors, topographic organization is an important principle also found in the human brain and has been used in various forms by research related to computational neuroscience. Integrating this idea with VAEs may foster future research in that direction.
- However, no results have been presented that show the advantage of a spatially topographic (i.e., L=0) latent representation, e.g. in terms of the sample quality or model interpretability/controllability. So this model is most likely of less interest to researchers not interested in topography itself.
- The temporal model (L > 0) has shown appealing results regarding the completion of sequences of complex transformations. Compared to related works, e.g. on disentanglement, this has important limitations: The learned transformations have to be presented as sequences during training. More importantly, only image sequences can be encoded, and samples can only be transformed by the transformation observed in the input sequence. So the applicability of the learned equivariance of the latent space is very narrow.

**Time Spent Reviewing:**

6

---

> ### Author Response · Authors · 2021-08-10
> **Initial Response to Reviewer P3ig**
>
> **Summary**
>
> We thank the reviewer for their thorough review and insightful questions which have helped us improve our work. We appreciate that the reviewer finds our work original and that it has potential to foster future work on topographic organization. In the following, we describe the structure of the representation learned by the temporally coherent TVAE in more detail, we provide samples from our model, and provide a baseline method for approximating the encoding of $t$ without sequence inputs.
>
> **Samples to Validate the Distribution of Latent Variables**
>
> We agree that samples from our model would be valuable to validate the distribution of latent variables after training. We thus generated the following images by drawing random samples $\mathbf{t}\sim TPoT$, and passing them through the generative model $g_{\theta}$ as trained in Section 6.2: ([LINK](https://raw.githubusercontent.com/ReallyAnonNeurips2021/TopographicVAE/rebuttal/tvae/figures/TVAE_2D_Samples.png)). As can be seen, the samples are qualitatively similar to real MNIST digits (accounting for the relatively low capacity of the model), implying that latent variables do indeed approximately follow the intended TPoT distribution after training. In the remainder of this response we will describe a procedure for sampling from the temporally coherent TVAE (L>0) and observe similar results.
>
> **Structure of the Temporally Coherent TVAE Representation**
>
> To answer the reviewer’s question about how the model represents constants in sequences, we now describe the structure of the learned TVAE representation in more detail. As noted, the TVAE is different from existing ‘disentanglement’ models (e.g. $\beta$-VAE) where input transformations are typically represented by the transformation of a single latent dimension. The TVAE instead represents transformations in a distributed manner, in the same vein as the operators described by [Bouchacourt et al. 2021](https://arxiv.org/abs/2102.05623). In fact, as additionally described in our response to Reviewer GHqG, the representation learned by the TVAE can most intuitively be described through a comparison with the group equivariant neural networks of [Cohen & Welling 2016](https://arxiv.org/abs/1602.07576). In such models, a transformation in input space corresponds to a roll through the group dimension of the representation (i.e. the capsule dimension). One could then see the constants of a sequence as being represented by the *set* of $L_p$ norms of all capsules, which are unaltered by the capsule Roll. This capsule-norm is called the ‘capsule activation’ in the [Capsule Networks of Sabour et al. 2017](https://arxiv.org/abs/1710.09829).
>
> Following this, we believe the same intuition can help describe how the $t$ representation of a single image changes when it is part of different input sequences. Since each capsule represents a feature *and* a transformation simultaneously, the set of capsule-norms for an image in two different sequences will likely be mostly distinct. An interesting way to interpret this is that the $u$ encoder is learning ‘constraints’ rather than features (similar to [Hinton & Teh 2013](https://arxiv.org/abs/1301.2278)): A capsule is largely active only when most constraints are satisfied (most $u$ in a capsule are < $\epsilon$), and constraint violations (i.e. misaligned sequence elements) decrease the norm of the capsule. In this view, one could see that a single image would have the same $z_l$ and $u_l$ encodings in two sequences, but the $u_{l\pm\delta}$ of the neighboring sequence elements would then decrease the norms of all capsules except those corresponding to the observed transformation.
>
> **Encoding/Sampling without Sequences**
>
> With respect to the reviewer's concern regarding the limitation that the TVAE requires input sequences for encoding the $t$ variable, we agree that this is a limitation of the model, but we also note that solutions to this limitation are not inconceivable. To exemplify this, we provide a simple method for approximating the $t$ variable from a single $z_l$ & $u_l$, and validate the approximation experimentally.
>
> Assisted by the above ‘constraint-learning’ interpretation, we hypothesize that the rolling operation of U variables in Equation 9 serves to align all active/inactive U’s within each capsule to prevent all capsules from being deactivated. This process is depicted graphically in our proposed update to Figure 1 ([LINK](https://raw.githubusercontent.com/ReallyAnonNeurips2021/TopographicVAE/rebuttal/tvae/figures/Figure_1.png)) — note how the activations (shown as light and dark squares) are rolled forward and backwards in the denominator corresponding to their timestep (denoted by increasing length curved arrows). Thus, we propose the denominator of $t_l$ could be simply approximated as $2L$ times a single $W_{\delta} u^2_l$.  Formally:
>
> $t_l=\frac{z_l-\mu}{\sqrt{W\left[u_{l+L}^2;\cdots;u_{l-L}^2\right]}}\approx\frac{z_l-\mu}{\sqrt{2L\cdot W_{\delta} u^2_l}}$
>
> To quantify this approximation, we provide the following figure which shows the input image (top), the reconstruction using the sequence-based $t_l$ (middle), and the reconstruction using this approximation (bottom): ([LINK](https://raw.githubusercontent.com/ReallyAnonNeurips2021/TopographicVAE/rebuttal/tvae/figures/T_Approximation_Recon.png)). As can be seen, the approximation is imperfect, sometimes altering properties of the input, but generally preserves the majority of qualities
>
> We additionally show examples of samples from the model, constructed with this approximation, here ([LINK](https://raw.githubusercontent.com/ReallyAnonNeurips2021/TopographicVAE/rebuttal/tvae/figures/Samples_T_Approx.png)) and find them to be qualitatively similar to true training examples. Further, we see that such sampled $t$ also inherently correspond to sampled transformations, as defined by the total capsule activation set. We can investigate these transformations by performing capsule rolls on the sampled $t$, and observe transformations similar to those in training ([LINK](https://raw.githubusercontent.com/ReallyAnonNeurips2021/TopographicVAE/rebuttal/tvae/figures/Samples_Traversal_Lhalf.png)), where each row is constructed by a capsule roll. Although it is clear that this is a rough approximation, we provide it as evidence that future work will likely be able to ameliorate this limitation more fully. Such extensions could bridge the gap with existing disentanglement literature, extending the general applicability of the model.
>
> **Impact of Latent Dimensionality**
>
> To address the reviewer’s question in regards to how the number and dimensionality of capsules affects the performance of the model, we have performed additional experiments with 9 and 36 capsules, and capsules of dimensions 12 and 30. The test set log-likelihood of these models can be found in the table below:
>
> | | TVAE $L = \frac{5}{36}S$ | VAE $L=0$ |
> |--|-|--|
> | 9 Caps x 18 Dim | -193.6 $\pm$ 0.5 |  -217.3 $\pm$ 0.5 |
> | 36 Caps x 18 Dim |  -189.8 $\pm$ 1.2 | -204.1 $\pm$ 0.4 |
> | 18 Caps x 12 Dim | -191.3 $\pm$ 1.8 |  -211.8 $\pm$ 0.9 |
> | 18 Caps x 30 Dim | -189.7 $\pm$ 0.53 |  -205.3 $\pm$ 0.6  |
>
> We observe only a marginal effect from changing the dimensionality of the latent representation, and this effect appears consistent across the TVAE and baseline VAE. Furthermore, we observe that the choice of capsule dimensionality is fairly robust and need not exactly match the length of the sequence (18) to obtain strong performance. We provide examples of capsule traversals from such models here ([LINK](https://raw.githubusercontent.com/ReallyAnonNeurips2021/TopographicVAE/rebuttal/tvae/figures/Traversals_Changing_Dim.png)), and observe learned transformations qualitatively similar to those from the $CapDim = S$ models.
>
> **Rolling Capsules Individually**
>
> As mentioned above, the structure of the TVAE representation is such that the latent operator is distributed across all capsules for an observed transformation. Therefore, we do not expect rolling the latent representation along a single capsule to produce meaningful transformations. We provide examples of such single capsule rolls for the $L=\frac{13}{36}$ TVAE in the following figure: ([LINK](https://raw.githubusercontent.com/ReallyAnonNeurips2021/TopographicVAE/rebuttal/tvae/figures/Single_Cap_Travs.png)). In the figure, the top row shows the sequence produced by rolling all capsules simultaneously; the remaining rows (i=0-17) show the sequence produced by rolling the corresponding capsule $i$ individually. As can be seen, in order to reproduce the full observed input sequence, it is required to roll all capsules simultaneously.
>
> **TVAE without Temporal Coherence (L=0)**
>
> In regards to the value of topographically organized features without temporal coherence, we have to politely disagree with the reviewer in their assumption that such models would not be of interest to researchers without further demonstration. As described in our response to Reviewer GHqG, a number of existing works demonstrate the potential advantages of topographic representations for learning invariant features in both unsupervised ([Le et al. 2011](http://ai.stanford.edu/~quocle/LeZouYeungNg11.pdf), [Kohonen et al. 1997](https://ieeexplore.ieee.org/abstract/document/6796177)) and supervised ([Kavaukcuoglu et al. 2009](http://yann.lecun.com/exdb/publis/pdf/koray-cvpr-09.pdf)) settings, where objective metrics such as classification accuracy have been improved. Since these benefits were not the primary motivation of this work, (our primary motivation being a link between topography and equivariance), we thus did not explore them experimentally. However, we believe our proposed model is a more principled training method for achieving the same topographic organization that has been the goal of significant prior work, and we thus find it to have non-trivial value in a diversity of research communities.

---

> > ### Comment · Reviewer_P3ig · 2021-09-01
> > **Re: Initial Response to Reviewer P3ig**
> >
> > Thank you for your detailed response. In my view, the additional results (also those prepared for the other reviewers) help a lot to understand representation learned by the model. Therefore I will increase my rating of the paper.
> >
> > Overall I see this paper as a borderline case. The fusion of ideas from topographic ICA and VAEs is interesting, novel and technically sound to me. As you stated, your primary focus however is the link between topography and equivariance. Including the additional results and responses to all reviewers, this work shows promising results in that direction. However I still have concerns regarding the limitations of the approach, due to the entanglement of the representation of the image and the sequence it is part of. Moreover, the transformations have to be given rather explicitly during training. The results concerning the approximation of the encoding without a sequence look indeed promising, as well as the capabilities of the model to generalize to unseen combinations of transformations. A more thorough treatment of these aspects would improve the generalizability and applicability of the model and further strengthen the paper in my view.

---

### Official Review · Reviewer_GHqG · 2021-07-18

**Rating:** 6
**Confidence:** 4

**Summary:**

**Summary**:
This paper first introduces the topographical VAE (TVAE): which is a latent variable model with a topographical product of student-t prior, and is trained with the variational autoencoding objective. They further show that the topographic organisation in TVAE can be leveraged to learn a basis of equivariant capsules, by using a neighbourhood structure which induces disjoint topologies, and by encouraging “shifting temporal coherence”. The authors test their models on MNIST and synthetic sequential data.

**Contributions**:
- This paper shows that the existing topographical product of student’s-t model can be constructed from a set of independent standard Gaussian random variables, and can be trained efficiently through variational inference.
- This paper discusses the disjoint neighbourhood structure in TVAE where each neighbourhood can be seen as a capsule.
- This paper shows that, when taking sequence as inputs, encouraging shift temporal coherence could lead to unsupervised learning of equivariance in capsules.
- In preliminary experiments, they demonstrate that TVAE learns to organise its activations according to salient characteristics on MNIST. They further show that, by incorporating topographic organisation over time (temporal coherence), they can learn equivariant capsules from synthetic sequential data.


**Ethical Concerns:**

I have no ethical concerns regarding this work.

**Limitations And Societal Impact:**

**Limitations**:
Below are a few of my suggestions:
- This paper should include more empirical and theoretical analysis of TVAEs (without temporal coherence) on still images.
- This paper should probably explore how to learn equivariant capsules from sequential data where the underlying transformations can change over time (or being continuous). If this is hard, what is the authors’ opinion on how we can potentially apply this approach to real-world sequential data?

**Societal Impact**:
I think the authors already have a good discussion of potential societal impact.


**Main Review:**

**Originality**:
Although topographical product of student’s-t model is already known, it is previously trained with techniques such as contrastive divergence. This paper shows that the model can be constructed from a set of independent standard Gaussian random variables, and can be trained efficiently through variational inference, therefore proposing topographical VAE (TVAE).

This paper also discussed topographical organisation over time (temporal coherence), and how to incorporate it into TVAE to learn approximately equivariant capsules from sequential data in an unsupervised manner.

**Quality**:
This paper contains many interesting ideas that are worthwhile to investigate further, including VAE training of topographical product of student’s-t model, the topographical organisation over time, unsupervised learning of equivariant capsules. However, because the authors want to put too many things in one work, the discussions or experiments for each part are inadequate in my opinion. For example, what’s the merits of having VAE training rather than contrastive divergence for TVAE? Why would people want to use TVAE rather than standard VAE for various applications? I can only get rough ideas of the properties of TVAE from the preliminary experiments on MNIST, but then the authors move on to discuss learning equivariant capsules. However, for learning equivariant capsules, the discussion is also inadequate. What would happen if the underlying transformations at each time step are different? Moreover, by rolling variables at each time step, the authors are using a permutation representation of group actions. But why can’t the authors use a simpler matrix representation, so that the dimensions of the learned representations can be greatly reduced?

I would like to also highlight my biggest concern of this paper below: If the roll operator is predefined and it rolls variables in each capsule by $1$ unit at each time step, is it suggesting that the input sequences should also have an identical transformation at each time step? Could the model still learn equivariance, if the sequential data is a video of a dropping ball (so not constant speed, but with acceleration)? If it is true that this model requires a constant underlying transformation for each sequence, how can the model potentially be applied to real-world problems in the future when such sequential data is almost never available unless you have control over the data generating process. (I might be mistaken about this part, so I would appreciate it if the authors could explain.)

**Clarity**:
The paper is well written and the authors clearly have already put some effort into presenting their ideas. Below are some minor suggestions:
- In Figure 1, because some variables are omitted, it is not so obvious that this is actually rolling. Maybe draw fewer variables but don’t use “…”?
- Line 120-122, Using TPoT as an example before introducing it does not help.
- Section 4.5.2, when talking about equivariance, maybe the authors should specify group actions in both the input and output space.

**Significance**:
This paper presents several interesting ideas including VAE training of topographical product of student’s-t model, the topographical organisation over time, unsupervised learning of equivariant capsules. These ideas can potentially inspire other works in this area. However, I would also like to point out that there are inadequate demonstrations of properties and merits of these approaches, and there should be more ablation study for complexities in the proposed models, before other people can confidently build on top of them.

In general, this paper presents several interesting ideas, even though I still have the concerns listed above. I think it is a borderline case, so I will be ready to adjust my score and will carefully read the authors’ responses.


**Time Spent Reviewing:**

6

---

> ### Author Response · Authors · 2021-08-10
> **Initial Response to Reviewer GHqG**
>
> **Summary**
>
> We thank the reviewer for their careful review and concise list of our contributions. We appreciate that they find our ideas worthy of further investigation, and that they acknowledge the significant potential of the work to inspire other work in the area. Below we provide further discussion and experiments including training the temporally coherent TVAE on sequences with ‘accelerating’ transformations.
>
> **Merits of the VAE framework for Topographic Generative Models**
>
> First, we would like to address the reviewer’s question regarding the merits of the newly introduced VAE training compared with existing methods. As mentioned in ([Osindero 2004](https://arxiv.org/abs/2011.03535), and [Du & Mordatch 2019](https://arxiv.org/abs/1903.08689)), contrastive divergence is challenging to scale to deep nonlinear networks due to instabilities in training and slow mixing, generally limiting the flexibility and scalability of the trained architectures. Since flexible deep neural network encoders and decoders are standard in VAEs, we believe our formulation alleviates many of these limitations and allows exploration of this new class of generative models. Further, given the recently discovered relationship between VAEs and Nonlinear ICA ([Khemakhem et al. 2019](https://arxiv.org/abs/1907.04809)), we foresee a possibility to achieve provable identifiability in this framework which may be of interest to those studying causality.
>
> **TVAE without Temporal Coherence**
>
> Second, we would like to address the suggested limitation pertaining to a limited exploration of the TVAE without temporal coherence. While we certainly agree that further exploration of the TVAE without temporal coherence is likely of significant interest to both the generative modeling and computational neuroscience communities, this direction was not the primary motivation of our work. Our primary purpose in developing the TVAE was to achieve topographic organization in a deep neural network generative model, and thereby demonstrate a connection between topographic organization and existing group equivariant convolutional neural networks ([Cohen & Welling 2016](https://arxiv.org/abs/1602.07576)). We believe that a number of existing works demonstrate the potential advantages of topographic representations for learning invariant features in both unsupervised ([Le et al. 2011](http://ai.stanford.edu/~quocle/LeZouYeungNg11.pdf), [Kohonen et al. 1997](https://ieeexplore.ieee.org/abstract/document/6796177)) and supervised ([Kavaukcuoglu et al. 2009](http://yann.lecun.com/exdb/publis/pdf/koray-cvpr-09.pdf)) settings, achieving demonstrably invariant features and improved classification accuracy. Such prior work provides a clear blueprint for how the TVAE representations could be leveraged in future work for downstream tasks. The experiment in Section 6.2, Figure 3 was therefore not intended to fully analyze the capabilities of the TVAE, but rather to give a visual representation of topographic organization for a single time step to facilitate understanding of the overall model. We will clarify this in the paper and discuss this in regards to future work and limitations.
>
> **Sequences with Accelerating Transformations**
>
> To address the reviewers chief concern, we have performed an additional set of experiments where the observed transformation is *not* identical between timesteps. Explicitly, the following experiments are performed with accelerating transformations where the transformation between timesteps is proportional to $4t + t^2$ for timestep $t$. (e.g. rotation/hue angles {0 5 12 21 32 45 60 77 96 117 140 165 192 221 252 285 320 357}, or scales {0.60 0.61 0.62 0.64 0.66 0.68 0.71 0.74 0.78 0.82 0.86 0.90 0.95 1.01 1.07 1.13 1.19 1.26}). The capsule traversals for TVAE models trained on this dataset are provided here: ([LINK](https://raw.githubusercontent.com/ReallyAnonNeurips2021/TopographicVAE/rebuttal/tvae/figures/Accelerating_Transforms.png)). As can be seen, the model is able to learn such transformations equally as well as for constant velocity transformations. Further, the likelihood of the TVAE on this data (-175.96) is still higher than that of the VAE (-184.87). Intuitively, this performance can be explained by a comparison of the TVAE with a group equivariant convolutional neural network. Each step of the transformation can be seen as a group element, and thus weights can be shared (constrained by such group elements) regardless of their relative distances (i.e. transformed copies of weights can be maintained within a capsule regardless of the intermediate transformations). The main constraint on the underlying transformations in our setting is that they are deterministic, and that there exist some finite set of transformations which operate on the training set, which we additionally assume to operate on the test set -- in such a scenario, the TVAE model should be applicable.
>
> **Choice of Roll as Latent Operator**
>
> To answer the reviewer’s question regarding why we chose a permutation representation of group actions in the latent space: this was mainly chosen to match the latent operator of group equivariant neural networks and facilitate this connection intuitively. In a group CNN, the activations permute one step within the group dimension (i.e. capsule dimension) for the action of the group on the input space. Through our Capsule Correlation metric (CapCorr) in Table 2 we have quantitatively shown that the TVAE learns a representation with the same latent cyclic permutation operator. Further, by inspecting the weights of a TVAE with linear encoders and decoder, we can explicitly observe the transformed copies of weights being learned, as would be analytically constrained in a group equivariant CNN (See images of the decoder weights here ([LINK](https://raw.githubusercontent.com/ReallyAnonNeurips2021/TopographicVAE/rebuttal/tvae/figures/Linear_Decoder_Weights.png)). Each square image displays the weight vector corresponding to a single $t$ activation, and each row denotes a capsule.). We will add a further comparison between our model and group equivariant CNNs in the appendix. Despite our intentions with the current construction, we do agree with the reviewer that alternative representations of the group action in latent space could greatly reduce the learned representation dimensionality. In Appendix C.1, we have described one such possible alternative latent operator (a ‘partial roll’), which we have observed to work equally well in practice. We additionally believe other simpler matrix representations could be valuable directions for future research.
>
> **Clarity**
>
> Regarding clarity, we will certainly heed the reviewers suggestions, and appreciate their notes. Additionally, please see our updated Figure 1 here ([LINK](https://raw.githubusercontent.com/ReallyAnonNeurips2021/TopographicVAE/rebuttal/tvae/figures/Figure_1.png)), which hopefully addresses the listed concerns.

---

> > ### Comment · Reviewer_GHqG · 2021-08-26
> > **Post-Rebuttal Discussion**
> >
> > I would like to thank the authors for their detailed response. I am mostly satisfied with comments regarding the merits of VAE training, and why temporal coherence is an important part of this paper. And I think it is also ok to only explore roll operator rather than other group representations in this work, considering there is already enough content.
> >
> > But I am not sure if I fully understand the experiments on sequences with accelerating transformations. Is it correct that when rolling the capsule dimension with a constant speed (e.g. 1 unit at each time step), the generated sequence will show accelerating rotations/hue or scale changes (e.g. increasing rotation angles at each time step)?  From my understanding, equivariance means that a transformation on the input space corresponds to a fixed transformation on the output space. So 1 unit rolling should correspond to a fixed rotation angle, even though the model is trained on sequences with accelerating transformations. Could the authors clarify this? Thanks.

---

> > > ### Author Response · Authors · 2021-08-30
> > > **Response to Reviewer GHqG’s Discussion Question**
> > >
> > > We thank the reviewer for their further discussion and allowing us the opportunity to provide additional explanation. The reviewer is correct that an equivariant map is one in which a transformation of the input corresponds to a fixed transformation of the output, i.e. $\Gamma_{\rho}[f(\mathbf{x})] = f(\tau_{\rho}[\mathbf{x}])$. They are also correct that, as demonstrated, when training on accelerating transformations, rolling the capsule dimension with a constant speed yields a generated sequence with accelerating transformations. More formally, we could say that the set of transformations which act on the input space is simply a set of elements {$\tau_0, \tau_1, \tau_2, \tau_3, \ldots $} (e.g. rotation angles {$0, 5, 12, 21, \ldots$}), and the corresponding representations of these transformations in output space are {$\Gamma_0, \Gamma_1, \Gamma_2, \Gamma_{3}, \ldots$} $=$ {$\mathrm{Roll_0}, \mathrm{Roll_1}, \mathrm{Roll_2},  \mathrm{Roll_3}, \ldots$}. Each input feature thus has a canonical orientation (given by $\tau_0$), and rotating that feature by $\tau_{\rho}$ corresponds to a transformation in capsule space of $\Gamma_{\rho}$, or a capsule roll of $\rho$ units (i.e. $\Gamma_{\rho} = \mathrm{Roll_{\rho}}$). Thus, each input transformation does indeed have a fixed corresponding output transformation, but it is fixed with respect to the index of the group element $\rho$, and therefore implicitly also fixed with respect to the ‘canonical’ pose of the feature (element $\rho=0$). In other words, for a capsule roll of one unit, the transformation observed in the input space is a function of the current state of the capsule activation (in some sense the ‘currently estimated pose’).
> > >
> > > Therefore, the observed transformation (i.e. the action of $\tau_\rho$, or the rotation angle) need not be a linear (or even continuous) function of the index $\rho$. In the context of our proposed model, smooth continuous transformations are beneficial since we can then leverage coherence within the internal capsule dimensions (values of $K>1$), however this need not always be the case. As an example of this, we ran the following experiment: training the TVAE with a fixed random sequence of angles, hues, and scales. You can find the capsule traversals here: ([LINK](https://raw.githubusercontent.com/ReallyAnonNeurips2021/TopographicVAE/rebuttal/tvae/figures/Random_Transforms.png)). As can be seen, the model is again able to learn this apparently random pattern of group elements (albeit slightly less easily), agreeing with the above intuition.
> > >
> > > If it is desired that $\mathrm{Roll}$ing $\delta$ units corresponds explicitly to a change of $\delta$ units of an observed quantity, rather than $\delta$ group indices, we could foresee this to be possible by inferring the value of the observed quantity between subsequent observations, and explicitly transforming the representation (e.g. rolling the capsules) according to the differences in this quantity during training time. One way this could be achieved is through a method similar to the weakly supervised technique of [Bouchacourt et al., 2021](https://arxiv.org/abs/2102.05623), where the phase correlation technique of [Reddy \& Chatterji, 1996](https://ieeexplore.ieee.org/document/506761) is used to infer the change in pose between two observations.

---

> > > > ### Comment · Reviewer_GHqG · 2021-08-30
> > > > **Response to Authors' Comments**
> > > >
> > > > I really appreciate the authors' detailed comments and admire the effort they put into this rebuttal. I still think this paper contains many interesting ideas that are worthwhile to investigate further. But given the current state of this work, I would like to keep my original rating.
> > > >
> > > > As a final comment, I think this paper would benefit from having a rigorous statement of what they mean by learning equivariance (mathematically specify group actions on the input/output space and verity they are indeed group actions.), and under what condition can they learn such representation. It should also be explicitly discussed that the learned transformations may be a function of the current state. In fact, I conjecture this is unavoidable if only image sequences are given. Moreover, if the authors would like to draw connections with G-CNNs, they could also discuss the relationship (differences w.r.t. feature spaces, group actions, etc.) formally somewhere.

---

> > > > > ### Author Response · Authors · 2021-09-02
> > > > > **Response to Reviewer GHqG**
> > > > >
> > > > > We thank the reviewer for their continued engagement in discussion. We understand the reviewer’s concern and will certainly add formalism (mirroring our [previous response](https://openreview.net/forum?id=AVWROGUWpu&noteId=2-PRM8nIoL_)) in our final paper. However, given our model requires no notion of groups with respect to the learned transformations, we believe formalism with respect to the language of traditional Group Theory would be slightly misleading. Indeed, the ‘accelerating transformations’ discussed in this thread are almost certainly not groups (they are not closed under composition), and thus would be inappropriately described by such a formalism, and yet our model still learns them accurately. We aimed to make a connection with G-CNNs in Section 3.2 (titled ‘Group Equivariant Neural Networks’), discussing how the representation of a transformation in the output space of the G-CNN is analogous to a capsule roll, but we will more explicitly clarify this connection, and highlight that our model is rather approximately equivariant to potentially non-group transformations, in the final paper.
> > > > >
> > > > > Additionally, with respect to learned transformations being a function of the current state, we would like to note that this is only an abstract definition of equivariance as it is usually defined, and indeed this property can even be seen to exist in specific convolutional layers. For example, consider a convolutional layer where the center of the image has a stride of 1, while this stride increases as you approach the edge of the image (e.g. stride 3 within some distance from the border). Such a layer would have a more dense representation of the input towards the center of gaze, something like the fovea in the biological system. If you then consider how a translation on the input space corresponds to the change of the representation in the output space, you would see a greater change (a larger output translation) for translations which occur in the center of the image, rather than for translations towards the edge of the input image. Symmetrically then, for a given representation, if we perform a translation of an activation in output space, the associated transformation in the input space will then be a function of the original location (i.e. the ‘current state’) of said activation -- larger observed translations for activations on the edges of the feature map compared with those on the center. Therefore, we do not see this property as specific to our model, but rather a potentially valuable, and perhaps expected, byproduct of *learned* approximate equivariance.

---

> > > > > > ### Comment · Reviewer_GHqG · 2021-09-02
> > > > > > **Response to Authors**
> > > > > >
> > > > > > I would like to thank the authors for their clarification. It answers many of my previous questions regarding equivariance.
> > > > > >
> > > > > > Yes, I agree it would be better to stress that the definitions of equivariance are different when drawing connections with G-CNNs. I would appreciate it if the authors could provide references to the convolutional layers with changing strides. I am also curious about the following: When two sequences of the same object are present in the data, one has a constant speed, the other has an accelerating speed. What would the learned equivariance be like? i.e. when you do capsule traversal, will you observe accelerating transformations but with a reduced acceleration? It seems to me that, this approach effectively learns the orbits of transformations, but the correspondence between elements on the orbits can be arbitrary depending on the training data.

---

> > > > > > > ### Author Response · Authors · 2021-09-02
> > > > > > > **Response to Reviewer GHqG**
> > > > > > >
> > > > > > > We sincerely appreciate the reviewer’s interest in our work. Given the relative infancy of such approximate equivariance as a topic, we are unaware of any references where ‘output-state dependent’ transformations would be observed. Our aim was to provide the convolution example as a thought experiment demonstrating the occurrence of such a phenomenon without the added complexity of learned transformations.
> > > > > > >
> > > > > > > In regards to the reviewer’s proposed experiment (restated: suppose the dataset contains two sequences of the same object, transforming with different acceleration), we hypothesize that the model would learn roughly two different sets of features (two sets of capsules) which equivalently have two different rates of acceleration within their capsule dimensions. Thus a capsule traversal would always be able to decode a sequence which is coherent with the partial  input sequence by only activating the capsules which correspond to the observed transformation. We thus think the reviewer’s understanding of orbit-learning is on the right track, but must take into account the provided input sequence which determines the sequence reached through the capsule roll.  This process can be understood more thoroughly through the lens of constraint learning as described in our [response to reviewer P3iG](https://openreview.net/forum?id=AVWROGUWpu&noteId=EptjgnPiajl) -- the elements of the partial sequence which do not align with the transformation contained in a capsule will deactivate that capsule, and thus only the matching transformations will be decoded.
> > > > > > >
> > > > > > > We have performed a similar experiment early-on in our development of the model, where we randomly alter the speed of rotation for each sequence. Explicitly, considering only rotation transformations, each 18-element training sequence either features a total 360-degree rotation or a total 720-degree rotation (i.e. either rotating 20 or 40 degrees between steps). Therefore, during training, the model will see many of the same objects undergoing different but similar transformations. Please see the results of that experiment for a single layer fully-connected model here: including [capsule traversals](https://raw.githubusercontent.com/ReallyAnonNeurips2021/TopographicVAE/rebuttal/tvae/figures/RandomSpeed_Traversals.png), and [visualization of the decoder weights](https://raw.githubusercontent.com/ReallyAnonNeurips2021/TopographicVAE/rebuttal/tvae/figures/RandomSpeed_DecoderWeights.png). We observe exactly the above description, the capsule traversal is able to decode the sequence with the correct speed, and the decoder weights appear to be roughly segregated into two groups -- those which rotate 360-degrees within each capsule (each row), and those which rotate 720 degrees within each capsule. We have attempted to annotate the associated capsule distinction manually in the figure. The distinction in weight space is not exact since this represents a linear basis used to reconstruct the input, however we hope most viewers would agree some distinction is clearly visible.

---

### Official Review · Reviewer_h1aQ · 2021-07-19

**Rating:** 6
**Confidence:** 3

**Summary:**

The paper introduces a VAE that can model the spatial and temporal “coherence” of latent variables. The idea is to construct student’s-t distributions from initial Gaussian latents. These student’s-t distributions are made coherent by sharing some of the Gaussian latents that are used in their construction. These variables are then divided into blocks called capsules where the variables in different blocks do not share any Gaussian latent vars. Temporal coherence is achieved by applying a similar “sharing” of Gaussians in a moving time window. However, the paper also suggests “rolling” the variables inside a capsule. The paper’s claim is that this rolling of the variables leads to an equivariant representation. Both qualitative and quantitative experiments on variations of MNIST and dSprites demonstrate the ability of this model in learning representations that are spatially and temporally coherent.


**Limitations And Societal Impact:**

yes!

**Main Review:**

The main contribution of the paper is in proposing a mechanism for achieving coherence in VAEs. This idea is quite appealing and also supported by the experiments. However, the claim that topographic representations learned by the model is necessarily equivariant needs more work. First, note that equivariance is more strict than coherence, in the sense that each sequence should be produced through symmetry transformation. With this assumption, each sequence can be a one-dimensional submanifold of a Lie group. It is not clear how a combination of capsules each of which is equivariant to SO(2) can represent such arbitrary one-dimensional submanifolds. This undermines a key claim of the paper, which also appears in the title.

Let’s see the experimental setup: the symmetry group is a product of SO(2)s, and compatible with the rolling operation of the capsule (note that this is not general). Moreover, the “sequences” used in experiments seem to have been produced by only changing one property at a time, and the same is done for testing the model. Therefore, train and test sequences can be quite similar, and the benefit of learning equivariant representation, in generalizing to new transformations from the same group, is not validated.

Did I understand the model and your setup correctly?

Could you please further elaborate on the capsule traversal process? Do you roll ALL the capsules at the same time? If so would this be able to produce a range of different one-dimensional submanifolds of a group that shares parts of a sequence (even the group is SO(2)xSO(2))? As an example, in a sequence, you may translate a digit on different trajectories in the training data, but suppose all these trajectories share parts of the initial sequence. How does the traversal process work for this example, when you feed parts of the sequence that is shared in these sequences?

As for other aspects of the paper, the coverage of related literature is quite extensive and it shows a broad perspective in this general area. However, the presentation can be improved especially at the beginning; I had little idea about what the paper was trying to achieve until section 4.2. (this may be because I wasn’t familiar with the meaning of the keyword topography in the context of representation learning.) Also, the champion figure is difficult to interpret partly because the rolling operation is not clear in the example.

Minor:
Q vs q in Eq(12)
Missing reference on line 217
Measuring capsule metrics on baseline models doesn’t seem fair (lines 262-263 need clarification.)
Figure 3 is produced by using inputs that maximize the activation of certain units. Is it possible to generate a similar traversal figure by “translating” the posterior for a particular input?
Can you test the traversal procedure by “combining” transformations (e.g., rotation+change of color in Fig 4)?


**Time Spent Reviewing:**

5

---

> ### Author Response · Authors · 2021-08-10
> **Initial Response to Reviewer h1aQ**
>
> **Summary**
>
> We thank reviewer h1aQ for reviewing our paper and spending a considerable amount of time with their review. We appreciate that they find our ideas appealing and that they have helped us clarify our experimental setup. We believe they have largely understood the model and setup correctly, however we have a few clarifications which may help answer their questions and concerns. Finally, we provide additional experimental results including capsule traversals for "combined" transformations not observed in training.
>
> **Experimental Setup Details**
>
> First, we would like to more explicitly define our experimental setup as follows: The symmetry group for the initial state of each sequence is a direct product of discrete cyclic groups (or semi-groups). Explicitly, the discrete groups are defined by a set of fixed rotation angles, hue ‘angles’, and scales for MNIST (e.g. {0, 20, 40, …, 340}, {0, 20, 40, …, 340}, {0.6, 0.64, …, 1.26}), and for dSprites we similarly select a fixed subset of x-positions, y-positions, orientations and scales. The training ‘sequences’ are then produced by choosing one of the subgroups (i.e. one of the properties) and sequentially enumerating all remaining elements of the subgroup. The model is encouraged to become approximately equivariant with respect to the individual discrete cyclic subgroups and not explicitly with respect to the much larger group formed by the direct product. If we understand correctly, we believe this is a slightly different setup than that explained by the reviewer which considers continuous Lie groups, and an assumed equivariance with respect to the direct product group. We appreciate this specification and will modify the experimental section of the paper accordingly.
>
> **Extent of Learned Equivariance & Combined Transformations**
>
> Pursuant to the above experimental setup, we agree that the experiments in the paper do not validate the Topographic VAE’s ability to learn equivariant representations w.r.t. continuous Lie groups, the aforementioned larger direct product of subgroups, or to generalize learned equivariance properties to unobserved group elements. We will specify the approximate equivariance properties of our model more precisely in the final paper, and add a discussion of this in the limitations. However, we do believe that the presented experiments validate the TVAE’s ability to learn representations which are equivariant w.r.t. the observed discrete cyclic subgroups (such as the group of 20 degree rotations) for the group elements presented in training (as exemplified by Fig 4 & Table 2). Additionally, following the reviewer’s suggestion, we provide examples of capsule traversals for ‘combined’ transformations not seen in training here: ([LINK](https://raw.githubusercontent.com/ReallyAnonNeurips2021/TopographicVAE/rebuttal/tvae/figures/Combined_Transforms.png)). The traversals are produced by the trained model weights from Figure 4 (left), tested on combined color and rotation sequences applied to the MNIST test set. As can be seen, the model is able to complete the sequences with high accuracy despite never having seen such combined transformation sequences in training. We believe the coherence of these traversals with the true input sequence is a promising sign for the generalizability of the learned TVAE representations.
>
> **Capsule Traversals for Shared Input Sequences**
>
> Pertaining to capsule traversals; the reviewer is correct that we roll all capsules simultaneously to produce the $\mathbf{t}_l$ variable for timestep $l$. We agree that this is not the most natural formulation with which to model a direct product of groups, and believe it would be valuable to explore the multidimensional capsule extension described in Appendix C.3 in future work. Such a multi-dimensional capsule model would encode each subgroup as a separate dimension in the capsule ‘lattice’ (not to be confused with internal capsule dimensions corresponding to individual neurons). Multiple transformations could then be created for a single input by rolling an activation along the corresponding transformation dimension of the lattice.
>
> We hope the above discussion additionally helps answer the reviewer’s question regarding how the capsule traversal process would work for shared input sequences, but provide additional elaboration below for completeness:  As mentioned in the limitations, since we only learn 1-D circular capsules, the only transformation which is reachable through a capsule roll is that which is deterministically given by the partial input sequence. If the partial input sequence is identical between two full length sequences, the model would be unable to decode both potential outcomes simultaneously and would likely model only one or an interpolation of both. However, if there is any subset of the input sequence which differs and indicates the identity of the full sequence, the model should be able to learn to decode the correct sequence. We believe this is demonstrated minimally on both datasets where a single element (the initial element) is always shared between multiple potential sequences (defined by the randomly chosen subgroup), and the model is able to decode the correct sequence by considering the remainder of the partial input sequence.
>
> **2D Capsule Traversals (Alternative For Figure 3)**
>
> For Figure 3, it is indeed possible to generate a similar figure by performing a 2-D capsule roll for a particular input. Please see such an example here: ([LINK](https://raw.githubusercontent.com/ReallyAnonNeurips2021/TopographicVAE/rebuttal/tvae/figures/2D_Capsule_Roll.png)). We note that the relative organization of digit classes between the two methods is identical (i.e. we can observe the sequence 1,7,9,4,2 from left to right in both plots), but they are cyclically shifted copies of one-another due to the ambiguity of the initial ‘starting position’ (defined by the input image) for the 2-D capsule roll.
>
> **Clarity**
>
> Pertaining to clarity, we appreciate the reviewer’s feedback and we will aim to make the definition of topography and the goal of the paper more clear in the introduction. Additionally, please see our updated version of Figure 1 which we believe is improves upon the clarity of the roll operation: ([LINK](https://raw.githubusercontent.com/ReallyAnonNeurips2021/TopographicVAE/rebuttal/tvae/figures/Figure_1.png))
>
> **Capsule Metric Baselines**
>
> With respect to measuring the capsule metrics on the VAE baseline, we agree this is not a competitive comparison and intended it to be treated as equivalent to a ‘random baseline’ to provide bounds for the proposed metrics. We will add text to clarify this. As a more competitive baseline, we performed additional experiments where we explicitly trained the VAE baseline to minimize the equivariance loss $\mathcal{E}_{eq}$ in addition to maximizing the ELBO. In this case, we observe the resulting equivariance loss is (326.2 $\pm$ 0.5), nearly identical to the TVAE with $L=\frac{1}{2}$ (315.0 $\pm$ 0.6), implying both models are achieving a similar minimum possible value. However, from inspecting the capsule traversals of the supervised baseline, we see that no transformations have been learned within the capsule dimension, implying it has actually learned an invariant, rather than equivariant, capsule representation: ([LINK](https://raw.githubusercontent.com/ReallyAnonNeurips2021/TopographicVAE/rebuttal/tvae/figures/Supervised_Baseline_Traversals.png)). Although further tuning, or more complex constrained optimization techniques, may improve the performance of this supervised baseline, we believe this experiment demonstrates the difficulty of training with such additional supervised losses, and thereby highlights a benefit of the TVAE framework. We are happy to include these results in the paper if the reviewer finds them valuable.

---

### Author Response · Authors · 2021-08-10
**Summary of Initial Author Response**

We would like to thank all reviewers for their considerable time spent with our submission -- we are grateful to have received three thoughtful reviews which have all contributed to improving this work. Briefly, we would like to summarize our understanding of the reviewers' main concerns and highlight the key points of our responses below.

All reviewers commented on the clarity of Figure 1. In response, we have provided a proposed updated figure which more clearly depicts the capsule $\mathrm{Roll}$ operator and its relationship to temporal coherence ([LINK](https://raw.githubusercontent.com/ReallyAnonNeurips2021/TopographicVAE/rebuttal/tvae/figures/Figure_1.png)).

Reviewer h1aQ mentions concerns related to the extent of the equivariance learned by our model and its generalization capabilities. In response we have provided a more explicit definition of our experimental setup and specified more precisely the learned equivariance properties which we believe our experiments validate. Further, we perform the additional experiment suggested by the reviewer and obtain positive results, implying that the TVAE may be able to generalize to combinations of transformations despite only being trained with individually transforming properties in isolation ([see capsule traversals here](https://raw.githubusercontent.com/ReallyAnonNeurips2021/TopographicVAE/rebuttal/tvae/figures/Combined_Transforms.png)).

Both Reviewers GHqG and P3ig suggested the analysis of the TVAE without temporal coherence is limited and thus questioned the value of the model. In response, we note that there exists significant prior work which demonstrates the benefits of topographically organized features ([Le et al. 2011](http://ai.stanford.edu/~quocle/LeZouYeungNg11.pdf), [Kohonen et al. 1997](https://ieeexplore.ieee.org/abstract/document/6796177), [Kavaukcuoglu et al. 2009](http://yann.lecun.com/exdb/publis/pdf/koray-cvpr-09.pdf)), and we highlight that the primary purpose of our work, as outlined in the introduction, is to bridge the concepts of topographic organization and equivariance, thereby demonstrating a novel potential benefit of such organization. Thus, although we agree that the study of the benefits of the TVAE  with respect to static images is of great interest, we find it more appropriate to leave such exploration for future work.

Reviewer GHqG raises a concern related to the applicability of the model to real world sequential data where transformations are rarely as consistent as demonstrated in our experiments. In response, we provide additional experimental results demonstrating the ability of the model to learn ‘accelerating’ transformations which have varying degrees of change at each time step ([see capsule traversals here](https://raw.githubusercontent.com/ReallyAnonNeurips2021/TopographicVAE/rebuttal/tvae/figures/Accelerating_Transforms.png)). We provide an explanation for this ability through comparison to [group equivariant convolutional networks](https://arxiv.org/abs/1602.07576), and further provide a set of assumptions about the data which bound the applicability of the TVAE.

Reviewer P3ig raises several questions regarding the representation learned by the model, including how constants are represented in sequences, how the representation of an individual image changes between two sequences, and how the image would be transformed if only a single capsule was rolled. To answer these questions we again relate the model to the group equivariant CNN, and further provide an interpretation of the TVAE in terms of constraint-learning. To validate this description of the model, we use it to derive a simple estimator for the encoding of $t$ variables without sequences (addressing another primary concern of the reviewer), and experimentally demonstrate the accuracy of this estimate by direct comparison of reconstructions ([reconstruction from exact $t$ shown in the middle row, reconstruction from estimated $t$ in the bottom row](https://raw.githubusercontent.com/ReallyAnonNeurips2021/TopographicVAE/rebuttal/tvae/figures/T_Approximation_Recon.png)). We additionally use this approximation to provide [samples from the L>0 model](https://raw.githubusercontent.com/ReallyAnonNeurips2021/TopographicVAE/rebuttal/tvae/figures/Samples_T_Approx.png), and [sampled sequences generated by capsule traversal](https://raw.githubusercontent.com/ReallyAnonNeurips2021/TopographicVAE/rebuttal/tvae/figures/Samples_Traversal_Lhalf.png), addressing the reviewers concern about a lack of samples. We finally provide further targeted experiments to answer the reviewer’s specific questions related to the dimensionality of latent variables and the effect of [traversing individual capsules](https://raw.githubusercontent.com/ReallyAnonNeurips2021/TopographicVAE/rebuttal/tvae/figures/Single_Cap_Travs.png).

We thank all parties involved for review of our submission and invite further discussion of any topics which need additional clarification.

---

### Decision · Program_Chairs · 2021-09-27

**Decision:**

Accept (Poster)

**Comment:**

Solid work making an original contribution on the link between topographic representations (such as those found in the brain)  to equivariance learning.